# Cell lineage and cell cycling analyses of the 4d micromere using live imaging in the marine annelid *Platynereis dumerilii*

B Duygu Özpolat[1†]*, Mette Handberg-Thorsager[2], Michel Vervoort[1], Guillaume Balavoine[1]*

[1]Institut Jacques Monod, Paris, France; [2]Max Planck Institute of Molecular Cell Biology and Genetics, Dresden, Germany

**Abstract** Cell lineage, cell cycle, and cell fate are tightly associated in developmental processes, but in vivo studies at single-cell resolution showing the intricacies of these associations are rare due to technical limitations. In this study on the marine annelid *Platynereis dumerilii,* we investigated the lineage of the 4d micromere, using high-resolution long-term live imaging complemented with a live-cell cycle reporter. 4d is the origin of mesodermal lineages and the germline in many spiralians. We traced lineages at single-cell resolution within 4d and demonstrate that embryonic segmental mesoderm forms via teloblastic divisions, as in clitellate annelids. We also identified the precise cellular origins of the larval mesodermal posterior growth zone. We found that differentially-fated progeny of 4d (germline, segmental mesoderm, growth zone) display significantly different cell cycling. This work has evolutionary implications, sets up the foundation for functional studies in annelid stem cells, and presents newly established techniques for live imaging marine embryos.
DOI: https://doi.org/10.7554/eLife.30463.001

*For correspondence:
dozpolat@gmail.com (BDÖ);
guillaume.balavoine@ijm.fr (GB)

Present address: †Marine Biological Laboratory, Woods Hole, United States

Competing interests: The authors declare that no competing interests exist.

## Introduction

Development of a multicellular organism requires precise regulation of the cell cycle, which has crucial roles in cell lineage establishment, cell fate decisions, and maintenance of pluripotency. Many embryonic and post-embryonic developmental processes involve stem cells that repeatedly give rise to tissue founder cells while also self-renewing at each round of division. Cell cycle regulation defines the correct timing and pacing of divisions for generating the progenitor cells, as well as maintaining the potency of stem cells themselves (*Ables and Drummond-Barbosa, 2013*; *Barker, 2014*; *Yasugi and Nishimura, 2016*). Understanding the cell cycle characteristics of stem cells and the implications of cell cycle regulation requires a combined lineage tracing and live-cell cycle analysis approach at single-cell resolution. However, such high-resolution lineage tracing has been challenging in many traditional and emerging animal model systems, due to a wide range of practical limitations that spans from the inaccessibility of embryos or tissues of interest, to the unavailability of tools and techniques (reviewed in *Kretzschmar and Watt, 2012*). Therefore, in order to understand cycling behavior of stem cells and their progeny in vivo, studies are needed in organisms where continuous observations are feasible in intact individuals and tissues.

Spiralians are a group of Protostomes including segmented worms (Annelida), mollusks, ribbon worms, and flatworms, many of which undergo a stereotyped program of early cell divisions known as spiral cleavage (*Conklin, 1897*; *Henry, 2014*; *Lyons et al., 2012*; *Seaver, 2014*; *Wilson, 1892*). Blastomeres that arise from spiral cleavage show determinate cell fates and a strict correlation exists between cell division timing and cell fate determination. The 4d micromere (also called M for Mesoblast) is one of the micromeres created during the fourth spiral cleavage, and it is an evolutionarily

conserved blastomere across Spiralia (*Lambert, 2008*). In most spiralians, 4d gives rise to meso-derm, endoderm, and primordial germ cells (PGCs) (*Ackermann et al., 2005*; *Gline et al., 2011*; *Kang et al., 2002*; *Lyons et al., 2012*; *Meyer et al., 2010*; *Rebscher, 2014*; *Shimizu and Naka-moto, 2014*; *Swartz et al., 2008*). Within a given species, 4d follows stereotypical division patterns. Differences across species in the 4d lineage division program have been proposed as a mechanism for obtaining the diverse body plans present across spiralians (*Lyons et al., 2012*). Yet, despite the immense diversity within spiralians, only a few studies have looked into the 4d lineage in detail, and some of these studies could not employ high-resolution lineage tracing due to techniques used (*Fischer and Arendt, 2013*; *Gline et al., 2011*; *Gline et al., 2009*; *Goto et al., 1999b*; *Lyons et al., 2012*). In addition, very limited data are available detailing the relationship between the 4d lineage cell cycle characteristics and the resulting differences in cell fates (*Bissen, 1995*; *Bissen and Weis-blat, 1989*; *Smith and Weisblat, 1994*). Thus, the spiralian 4d lineage provides an exciting embry-onic stem cell model system for linking cell fate potency, cell lineage, cell cycle, and morphogenetic processes.

Annelids (segmented worms) are a large but understudied group of spiralians containing many species with broadcast-spawning, giving large numbers of relatively small, yolk-poor and translucent embryos and larvae, which are amenable to functional developmental studies. Annelids are charac-terized by repeated (metameric) body parts called segments (*Balavoine, 2014*). In clitellate annelids, a taxon including the leeches and earth worms (*Zrzavý et al., 2009*), founder cells of segmental tis-sues are generated during embryonic development by stem cells called teloblasts (*Anderson, 1973a*; *Devries, 1973*, *1972*; *Goto et al., 1999b*; *Penners, 1924*; *Weisblat and Shank-land, 1985*; *Zackson, 1982*). Specific teloblasts make specific tissue types. The 4d micromere is the originator of the M teloblasts (mesoteloblasts) that make the trunk mesoderm: the first division of 4d generates two bilaterally symmetric stem cells named *M*esoteloblast-*L*eft (ML) and *M*esotelo-blast-*R*ight (MR). ML and MR then make the left and right mesodermal *bands*, respectively, via a well-described division program (*Weisblat and Shankland, 1985*; *Zackson, 1982*): the teloblasts repeatedly divide asymmetrically to self-renew the ML/MR stem cells and to give rise to tissue pre-cursor cells (primary blast cells) through iterated divisions. Each primary blast cell (much smaller in size compared to the teloblasts they have split from) follows a stereotyped program of cell divisions with fixed fate, generating clonal regions of tissues in adjoining segments. Micromere 4d and its daughters ML and MR are evolutionarily conserved embryonic stem cells across spiralians (*Lam-bert, 2008*; *Lyons et al., 2012*). Their teloblastic nature in non-clitellate annelids has been sug-gested before (*Anderson, 1973b*; *Fischer and Arendt, 2013*), but direct evidence for teloblasts outside of clitellate annelids is still missing.

*Platynereis dumerilii* is a marine annelid (Errantia, Nereididae) suitable to address the questions outlined above. As an Errant 'Polychaete', *P. dumerilii* is phylogenetically distant from clitellates (*Struck et al., 2011*; *Weigert and Bleidorn, 2016*) and presumably much closer in anatomy to the last common ancestor of annelids (*Balavoine, 2014*). Based on comparative genome analyses, *P. dumerilii* has also been suggested to belong to a slow-evolving lineage, thus potentially bearing genomic ancestral features of annelids (*Raible et al., 2005*; *Raible and Arendt, 2004*). In addition, *P. dumerilii* has externally fertilized, relatively fast-developing, transparent embryos which can be injected for lineage tracing and can be cultured at the lab for the full life cycle (*Ackermann et al., 2005*; *Backfisch et al., 2014*). Embryos develop into free-swimming planktonic larvae in about 24 hr-post-fertilization (hpf). By 48 hpf, segmental organization starts to become apparent, mostly evi-dent by the repetition of paired bilateral bristle bundles (chaetae) on each segment (*Fischer et al., 2010*). At this stage, a mesodermal posterior growth zone (MPGZ) has formed anterior to the pre-sumptive pygidium (the posterior-most non-segmental region), juxtaposed with the four putative PGCs (pPGCs). The MPGZ and pPGCs, as a cell cluster, sit at the converging point of the left and right mesodermal bands, and both express Vasa mRNA and protein (*Rebscher et al., 2012*, *Rebscher et al., 2007*). The first two divisions of ML and MR in *P. dumerilii* give rise to the pPGCs (*Fischer and Arendt, 2013*). However, how the MPGZ and pPGCs end up next to each other, and the exact embryonic origin of the MPGZ within the 4d lineage are not yet known. Previous studies show that the mesodermal bands, and eventually the segmental mesoderm also originate from the 4d micromere in *P. dumerilii* (*Ackermann et al., 2005*; *Fischer and Arendt, 2013*), but whether the segmental mesoderm forms via stereotyped teloblastic divisions of primary blast cells (*à la* clitellate) is also unknown.

Here, using high-resolution live imaging techniques complemented with a live-cell cycle reporter we developed, we report an extensive analysis for the 4d lineage at single-cell resolution, and an investigation of cell cycling patterns of several lineages that originate from the 4d micromere. We have developed imaging techniques for both embryos and larvae that are easy to implement and can be applied to other annelids and spiralians, as well as other metazoans with ciliated larvae. We show that a pair of mesoteloblasts (ML and MR), similar to what has been observed in clitellate annelids, are active during *P. dumerilii* embryogenesis and that they give rise to the mesodermal derivatives and pPGCs via asymmetric cell divisions. A series of four contiguous primary blast cells produced on each side of the larva proliferate to produce mesodermal blocks that each correspond to a distinct larval hemisegment. We show that M cells, after having produced the four larval segments, undergo an abrupt transition in their cycling behavior and start dividing much more slowly and symmetrically. These final divisions of the mesoteloblasts give rise to cells that form the MPGZ in the early larvae. The MPGZ cells remain in contact with the pPGCs, which are produced earlier and arrested in G0/G1. Differences we observed in cell cycling patterns in differentially-fated lineages that originate from a single cell (4d) provide foundational information to start delineating the relationship between cell cycle regulation, cell lineage, and generation of different cell fates.

## Results

### Establishment of a work flow for long time-lapse live imaging and cell lineage tracing in marine embryos and larvae

Live imaging at single-cell resolution over extended time periods, and analysis of these 4D image datasets for determining cell lineages require overcoming technical barriers. In this work, we used a standard scanning confocal microscope, which allowed us to trace cells in a specific body region of the embryo. To this end, we overcame a number of specific difficulties, piecing together a complete workflow that could be applied to other marine embryos and larvae. For the immobilization of embryos and larvae, we mounted samples in a thin layer of low-melting sea-water agarose using glass-bottom dishes. After injection and a careful selection of normal developing embryos at the time of mounting, we observed normal development of all embryos and larvae in agarose. For embedding swimming stages, we developed a new deciliation protocol, allowing for permanent immobilization, even after cilia regrew (see Materials and methods).

Intense exposure to laser lights in a scanning confocal microscope caused phototoxicity problems, resulting in embryonic deaths. To solve this problem, each image stack (at a resolution of 512 × 512 pixels) was acquired in roughly 2 min and was followed by a recovery time of at least 5 min. This time seemed to allow for the natural elimination of the toxic free radicals created by the laser light, before reaching deleterious concentrations for the cells. In these conditions, we observed development of the embryos and larvae that are morphologically indistinguishable from control animals.

This time resolution (about 7 min, or longer depending on the stack thickness imaged), which helped limiting exposure of samples to the laser light, in turn limited the possibility of using automated lineage tracing. In addition, for datasets that have low time- or image-resolution, or embryos which have significantly variable cell sizes and nuclei, segmentation and lineage-tracing algorithms available via different platforms are currently not able to carry out an automated lineage analysis with high accuracy (*Ulman et al., 2017*). Consequently, lineage tracing in such samples still largely rely on manual curation. For determining cell lineages at single-cell resolution in *Platynereis dumerilii* (*P. dumerilii*), we used a combination of labeling techniques. In addition to the conventional nuclear and membrane labeling, we also used a live-cell cycle reporter. This added the necessary information to trace lineages accurately, making it easier to predict when a cell will divide and follow daughter cells after a division, without the need to increase time resolution and thus light exposure.

### Construction of a cell cycle reporter and analysis of its cycling patterns

In order to visualize cell cycle progression in live *P. dumerilii* embryos and larvae, we first constructed a fluorescent cell cycle reporter. Live-cell cycle reporters rely on fusion of a truncated cell cycle protein containing a degron motif (also called destruction box) to a fluorescent protein. As a result, the fluorescent protein becomes a visible reporter of the corresponding cell cycle phase, and gets

degraded when the endogenous cell cycle protein normally gets degraded. These cell cycle reporters are also called FUCCI (fluorescent ubiquitination-based cell cycle indicator) (*Sakaue-Sawano et al., 2008*; *Zielke and Edgar, 2015*). For developing a live-cell cycle reporter in *P. dumerilii*, we first identified in the *P. dumerilii* genome and transcriptomes the *cdt1* gene (*Figure 1A*), on which specific G1 phase reporters are based in human and zebrafish (*Sakaue-Sawano et al., 2008*; *Sugiyama et al., 2009*). Metazoan Cdt1 proteins contain a well-characterized degron motif called PIP box (Q-x-x-[I/L/M/V]-T-D-[F/Y]-[F/Y]-x-x-x-[R/K]) (*Havens and Walter, 2009*), which interacts with the ubiquitin ligase complex CRL4 pathway for degradation. The amino acid sequence of the degron in human Cdt1 (hCdt1) is **Q**RR**VTD**FFARR**R** and it is at position aa3-14. We found a conserved degron **Q**TS**VT**N**FF**ASR**K** at the same location in *P. dumerilii* amino acid sequence (*Figure 1—figure supplement 1*). We also searched for a second degradation peptide, the Cy motif (aa68-70 in hCdt1), that interacts with SCF E3 ligase for degradation (*Fujita, 2006*; *Nishitani et al., 2006*). This second degron, however, cannot be identified unambiguously in metazoan proteins (including *P. dumerilii*) outside of mammals. Assuming that the PIP box is the most important for *P. dumerilii* Cdt1 degradation, we then proceeded by fusing a large truncated Pdu-Cdt1 sequence (aa1-147) containing the PIP box degron, but excluding the other functional interaction or DNA-binding domains, to different fluorescent proteins (mVenus, mCherry, mKO2, mAG) (*Figure 1A*). The fused constructs were then cloned into pCS2+ expression vector, transcribed in vitro, and injected as mRNA into embryos at one-cell stage (see Materials and methods for details). Among all constructs tested, only mVenus-Cdt1(aa1-147) showed fluorescence located to the nucleus and clear cycling without producing phenotypic effects (*Figure 1B*). The other constructs which had different fluorescent proteins but the same Cdt1 peptide did not produce any fluorescent signal (a problem also encountered by other researchers while establishing cell cycle reporters in other systems).

To understand the temporal characteristics of cycling patterns of mVenus-Cdt1(aa1-147), we used live imaging and 5-ethynyl-2'-deoxyuridine (EdU) labeling assay. We first carried out a detailed time-point analysis for individual cells using live imaging. *HistoneH2A-mCherry* mRNA was co-injected with the *mVenus-cdt1(aa1-147)* mRNA for continuous nuclear labeling. The imaging was done in several embryos (n = 10, four replicates) but here we report results from a representative embryo (*Figure 1—video 1* and *Figure 1—video 2*). We found that mVenus-Cdt1(aa1-147) in *P. dumerilii* was present during G1 phase, and was then degraded, probably during S phase. In contrast to the widely used hCdt1 reporter, which is specific to G1, the *P. dumerilii* Cdt1 reporter started accumulating again shortly before mitosis probably in late G2 (*Figure 1B*, see cell 'ab' and then cell 'b' before mitosis). Consequently, mVenus signal was also present during mitosis (can be seen in *Figure 1—video 1*, *Figure 1—video 2*). Fluorescence was suddenly dispersed in the cytoplasm when the nuclear envelope broke down during mitosis (because the truncated chimeric protein, unlike the native Cdt1, does not bind to chromatin), and then quickly concentrated in the daughter nuclei when nuclear envelope reformed at the end of telophase. We observed contrasting cycling patterns in sister cells. As an example, nucleus 'a' retained the mVenus signal for more than an hour (*Figure 1B*, upper row), while nucleus 'b' kept signal for only about 8 min (*Figure 1B*, lower row). The cell 'b' eventually divided (at 52 min) while cell 'a' was still in G1/S phase. Similar cycling patterns were observed for other cycling cells in several videos we have obtained, thus establishing mVenus-Cdt1(aa1-147) as an efficient cycling marker.

Next, in order to determine more precisely when mVenus-Cdt1(aa1-147) was degraded, we used S-phase-labeling with EdU. Cdt1 protein is part of the DNA-replication complex, normally present at the beginning of S phase, but targeted for degradation during S phase in order to prevent re-replication (*Arias and Walter, 2006*; *Fujita, 2006*; *Liu et al., 2004*). Thus, we tested whether mVenus-Cdt1(aa1-147) signal overlapped with the EdU signal. Embryos injected with *mVenus-cdt1(aa1-147)* and *HistoneH2A-mCherry* mRNAs were incubated until 12 hr-post-fertilization (hpf), then treated with a very short pulse of EdU (3 min) and immediately fixed after quick rinsing (as EdU detection cannot be carried out live, specimens needed to be fixed). The EdU treatment was long enough to mark the S phase but too short for most of the cells to progress much further in the cell cycle. As a result, the vast majority of EdU(+) cells were expected to be in S phase at the time of sample fixation. We analyzed cell nuclei from six embryos (from two independent experiments) treated this way (*Figure 1C–D'''*). We observed that 15.6% (n = 51) of cells were positive for mVenus, mCherry and EdU, demonstrating that mVenus-Cdt1(aa1-147) was present during at least part of the S phase (*Figure 1D*). Only a few cells (2.1%, n = 7) were positive for mCherry and EdU, but not mVenus,

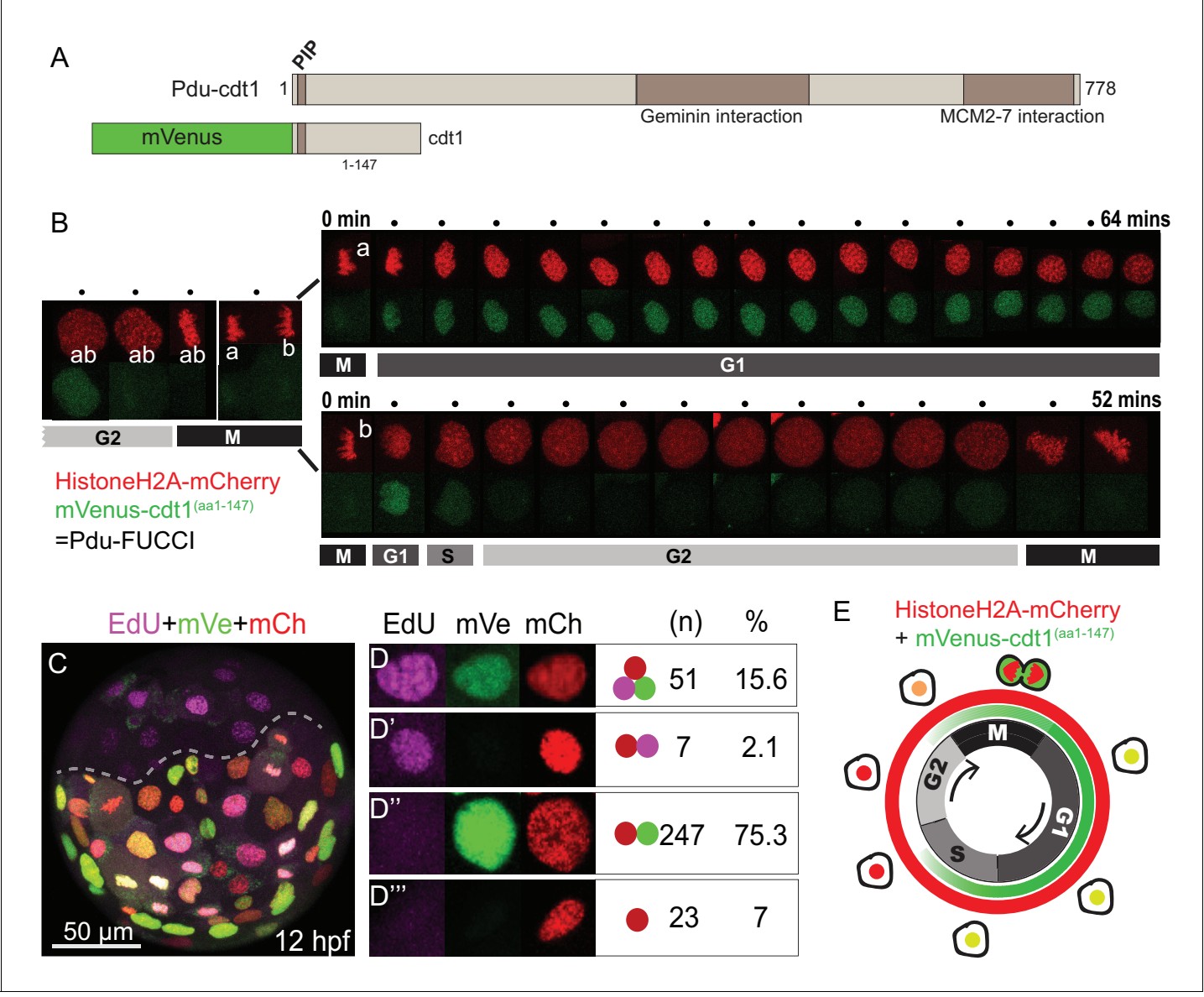

**Figure 1.** FUCCI cell cycle reporter for *Platynereis dumerilii*. (**A**) *P. dumerilii* Cdt1 complete amino acid sequence above, showing the domains including PIP destruction box (N terminal). Below: a representation of the fused cell cycle construct containing mVenus fluorescent protein and a truncated part of Pdu-Cdt1. mVenus is a yellow fluorescent protein but is shown in green color for convenience throughout the manuscript. (**B**) Frames (every 4 min) from time-lapse (*Figure 1—Video 1* and *Figure 1—Video 2*) showing division and cell cycling of an embryonic cell (labeled as 'ab') and its daughters ('a' and 'b'). Fluorescent channels are shown separately for HistoneH2A-mCherry (red - nuclear) and mVenus-Cdt1(aa1-147) (green - cycling nuclear). The sister cells show significantly different cycling patterns, as indicated by bars marking the different cell cycle phases (M/G1/S/G2). (**C**) Example of an embryo which was injected with mRNA at two-cell stage, EdU-incubated for 3 min at 12hpf, and fixed immediately. The part below the dashed line is the injected side. All channels are merged. Red: Histone2A-mCherry; Green: mVenus-Cdt1(aa1-147); Magenta: EdU. (**D–D'''**) Examples of cells that displayed different color combinations, and counts and percentages of cells based on data from six embryos (two independent experiments). (**E**) Cartoon summarizing the Pdu-FUCCI fluorescence patterns based on the time-lapse observations in B and EdU observations in D-D'''.

DOI: https://doi.org/10.7554/eLife.30463.002

The following video and figure supplements are available for figure 1:

**Figure supplement 1.** - Cdt1 amino acid alignments of destruction box sequences from different species.

DOI: https://doi.org/10.7554/eLife.30463.003

**Figure supplement 2.** Multiciliated cells that exit the cell cycle can be visualized using Pdu-FUCCI.

DOI: https://doi.org/10.7554/eLife.30463.004

**Figure 1—video 1.** Cell cycle reporter Pdu-FUCCI time-lapse in the early embryo (mCherry and mVenus).

DOI: https://doi.org/10.7554/eLife.30463.005

*Figure 1 continued on next page*

*Figure 1 continued*
**Figure 1—video 2.** Cell cycle reporter Pdu-FUCCI time-lapse in the early embryo (mVenus).
DOI: https://doi.org/10.7554/eLife.30463.006
**Figure 1—video 3.** Time-lapse of prototroch cells being born and exiting cell cycle.
DOI: https://doi.org/10.7554/eLife.30463.007
**Figure 1—Video 4.** Explanation of the onionization process, and display of all the onion layers.
DOI: https://doi.org/10.7554/eLife.30463.008

suggesting that mVenus-Cdt1(aa1-147) was degraded at some point during the S phase (*Figure 1D'*). Taking into account that the vast majority but not all Edu(+) cells are also mVenus(+), this strongly suggests that mVenus-Cdt1(aa1-147) is degraded toward the end of the S phase and is no longer present as the cells enter the G2 phase. The majority of cells (75.3%, n = 247) were found to be positive for mVenus and mCherry, but lacked EdU signal (*Figure 1D''*). These cells were either in G1 or late G2 phase. Finally, cells that were only mCherry(+) marking the early G2 phase made up a smaller percentage (7%, n = 23) of all cells analyzed (*Figure 1D'''*). We further confirmed that the cell cycle reporter is working properly by analyzing the ciliated cells that are known to be terminally differentiated in the annelid swimming larvae. These cells exit the cell cycle, and thus are in G0, no longer degrading Cdt1. As expected, we observed that they accumulated mVenus-Cdt1(aa1-147), once they stopped cycling. The mVenus signal reached much higher levels than in any cycling cell and remained at high levels in these fully differentiated multiciliated cells (*Figure 1—figure supplement 2*, *Figure 1—video 3* and *Figure 1—video 4*).

In *P. dumerilii,* mVenus-Cdt1(aa1-147) reporter is present during a larger portion of the cell cycle (*Figure 1E*). It is being degraded during the course of the S phase, similar to the vertebrate reporter, and it starts to accumulate again during the late G2 phase. This pattern reflects the behavior of endogenous Cdt1 degradation reported in other organisms: In human cells, Cdt1 starts getting degraded in the S phase, thus it is expected to be present at least partially during this phase (*Shiomi et al., 2014*). In embryonic stem cells, Cdt1 starts accumulating at the end of G2 phase, which is promoted by its interaction with Geminin (*Ballabeni et al., 2004*). Even though our mVenus-Cdt1(aa1-147) construct is not specific to the G1 phase, its cycling behavior is very pronounced as the protein is at barely detectable levels in late S and early G2 phases and therefore, together with HistoneH2A-mCherry, allows to follow cell cycle progression in living embryos. We call the two constructs (HistoneH2A-mCherry and mVenus-Cdt1(aa1-147)) together 'Pdu-FUCCI' in the rest of this manuscript. Next, we show our results of 4d micromere lineage analysis at single-cell resolution, via live imaging combining Pdu-FUCCI with membrane labeling. The layer of information into cell cycle progression provided by mVenus-Cdt1(aa1-147) was crucial in accurate manual curation of lineages in the experiments reported below. Because we could not increase the time resolution due to phototoxicity, cell cycle progression was used for predicting cell division, and differentiating the daughter cells from the surrounding cells.

## Live imaging the cell cycling and migration behavior of putative primordial germ cells

In *P. dumerilii*, the mesodermal 4d micromere (also called 'M') gives rise to two symmetrically positioned sister cells ML (**M**esoblast '**L**eft') and MR (**M**esoblast '**R**ight') by symmetric cell division (*Fischer and Arendt, 2013*; *Rebscher et al., 2012*, *Rebscher et al., 2007*). The following two highly asymmetric divisions of ML and MR give rise to four small cells (1ml/r and 2ml/r), which have been suggested to form the germline (putative primordial germ cells, pPGCs). Thus, the pPGCs are the earliest lineage to segregate from 4d in *P. dumerilii*. pPGCs become mitotically quiescent right after birth. As a result, they can be labeled with a short pulse of EdU around their time of birth (between 3 and 8 hpf), and be detected even days after EdU incorporation as they retain undiluted EdU while the neighboring somatic cells have largely diluted EdU through multiple mitoses (*Rebscher et al., 2012*). We confirmed that when embryos were treated between 5 and 7 hpf or 6 and 8 hpf, and raised to 24 hpf or 48 hpf all four pPGCs were labeled with EdU under our culturing conditions (5–7 hpf treatment shown in *Figure 2A* for samples raised until and fixed at 24 hpf, in *Figure 2B* for samples raised until and fixed at 48 hpf; 6–8 hpf data not shown). When embryos were treated with EdU between 7 and 9 hpf, four out of six larvae had only two EdU-positive pPGCs at 72 hpf (data not

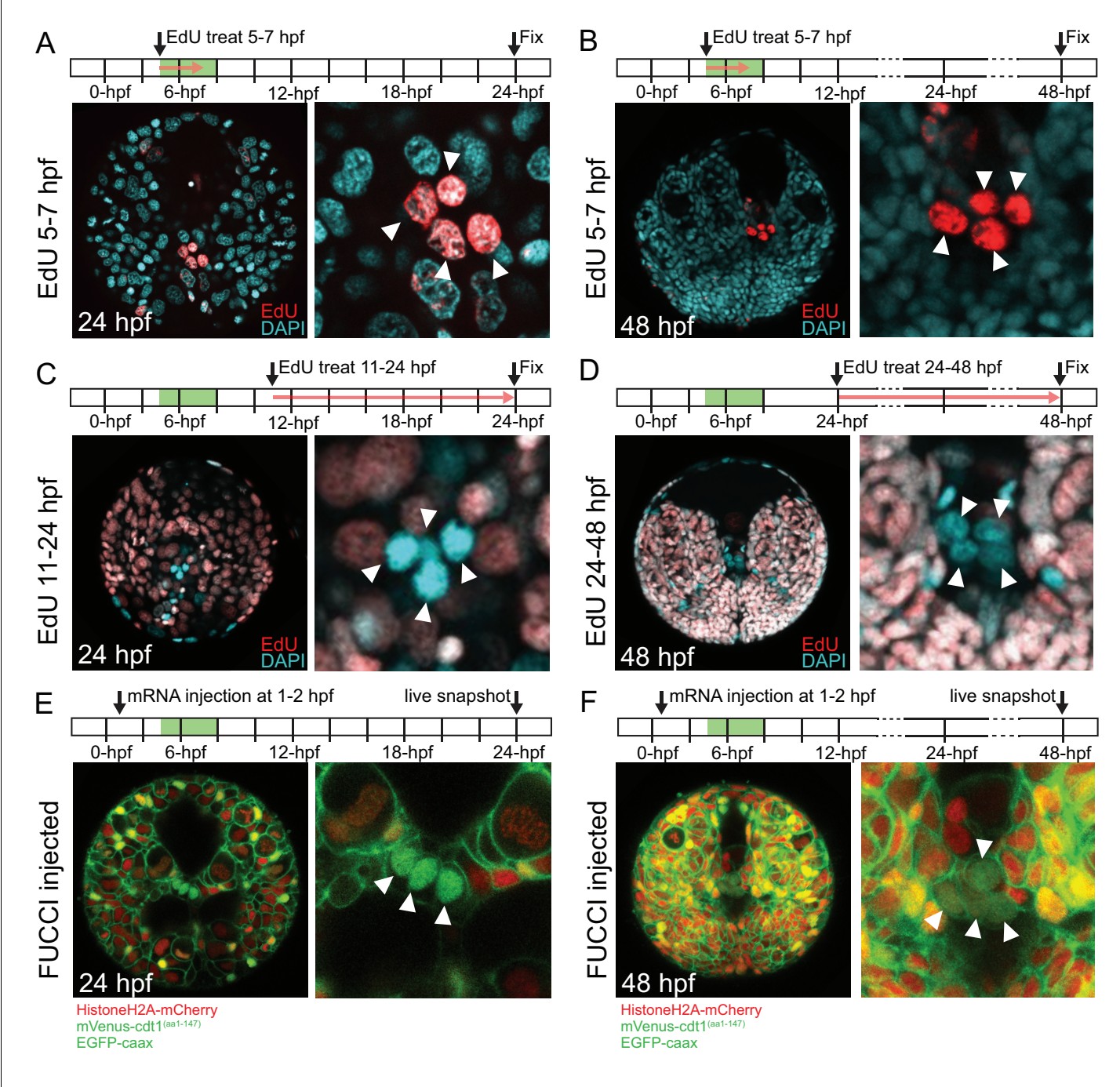

**Figure 2.** pPGCs do not incorporate EdU after they are born and retain mVenus-Cdt1(aa1-147) signal. pPGCs (arrowheads) incorporate and retain the EdU signal (red) when treated during the time they are born (5–7 hpf), and can be easily detected at 24 hpf (**A**) or 48 hpf (**B**) due to retention of EdU. However, they do not incorporate EdU, if treated after they are born (**C and D**). Live samples that were injected with Pdu-FUCCI (*HistoneH2A-mCherry, mVenus-cdt1(aa1-147)*) and *EGFP-caax* and imaged at 24 hpf (**E**) and 48 hpf (**F**) show that pPGCs have the nuclear mVenus signal (green), also suggesting that they have stopped cycling (see also *Figure 3*, *Figure 3—figure supplement 1*, and *Figure 7B*). Green bars in the timeline schemas show the time period in which pPGCs are born. Red arrows in **A–D** show the period of EdU treatment. At least n = 6 samples were imaged for each EdU assay, and a representative sample is shown.

DOI: https://doi.org/10.7554/eLife.30463.009

shown), indicating that pPGCs stop incorporating EdU around this time frame. In addition, to test whether pPGCs synthesized DNA after they were born in the next 2 days of development, we carried out EdU incorporation assays after pPGCs were born, between 11 and 24 hpf (*Figure 2C*) and 24 and 48 hpf (*Figure 2D*). We did not detect any pPGCs positive for EdU in the larvae (n > 16) in either of these treatments, confirming DNA synthesis does not take place during this time frame in the pPGCs.

We then analyzed the pPGCs for their cell cycle state using Pdu-FUCCI injection and live imaging. In this and following experiments, the injection cocktail included Pdu-FUCCI and an additional EGFP construct with a cell-membrane-localization signal (EGFP-caax) for visualization of individual cell shapes. First, we injected one-cell stage embryos with Pdu-FUCCI and EGFP-caax, and imaged the larvae live at 24 hpf and 48 hpf. We found that pPGCs were mVenus-Cdt1(aa1-147) positive at 24 hpf and 48 hpf, suggesting that they are arrested in G0/G1 (*Figure 2E and F*). Next, in samples injected with the same cocktail but only in the D quadrant at four-cell stage, we carried out high time-resolution (every 10 min) time-lapse imaging to determine if Pdu-FUCCI could allow tracking the pPGCs (*Figure 3*, *Figure 3—figure supplement 1*, *Figure 3—video 1*). We chose to inject the D quadrant only, as it eventually gives rise to the 4d lineage from which pPGCs originate. In addition, injecting only the D quadrant facilitated the observation of its descending lineages, because the rest of the embryo remained unlabeled. Starting from 8 hpf, injected embryos were imaged for about 12 hr, with 10 min intervals. Temperature conditions during live imaging were higher than 18°C (which is the usual culture temperature for *P. dumerilii*) (*Fischer et al., 2010*). Thus, development was accelerated during imaging allowing us to record a broader developmental period, while the sample had normal development (*Figure 3—video 2*). This imaging period corresponds to the stages from stereoblastula to mid-late trochophore (roughly 8 hpf to 38 hpf at 18°C) (See *Table 1* for time-points and corresponding stages at different temperatures). Using the high-resolution time-lapse video, we were able to track the pPGCs, and observed that mVenus signal persisted in pPGCs after they were born (*Figure 3A–H*, *Figure 3—figure supplement 1* for close-ups). Thus, in addition to differentiated cells such as ciliated cells, Pdu-FUCCI is a useful tool to trace the cells that are undifferentiated but arrested in G0/G1, such as the pPGCs.

We then investigated the migration behavior of pPGCs in the same imaging dataset from the sample injected in the D quadrant with Pdu-FUCCI and EGFP-caax mix. When live imaging was started, 4d cell had already divided once giving rise to the left and right mesoblasts (ML and MR), and ML and MR had divided twice giving rise to two pPGCs each (*Figure 3A–A''*, *Figure 3—figure supplement 2* shows the first two divisions of ML and MR in another sample). pPGCs at this stage (stereoblastula) were located on the external surface of the embryo (*Figure 3A'–A''*, *Figure 3—video 3*, *Figure 3—video 4*). However, soon after they were born pPGCs started moving toward each other, formed a cluster, and then moved toward the vegetal pole through epiboly (*Figure 3B'–D''*). The movement of pPGCs continued on the surface until 20 hpf (or about 5 hr of imaging – *Table 1*) (*Figure 3D'–D''*). Then they started to position internally (*Figure 3E'–G''*), eventually assuming a postero-ventral position forming a bridge between the left and right mesodermal bands which converge at the ventral pole (*Figure 3H–H''*, *2E and F*). This analysis establishes a framework for locating pPGCs through these developmental stages and allow for tracking these cells in experimental manipulation conditions such as drug treatment or cell ablation in future studies.

## Each mesodermal hemisegment is the progeny of individual blast cells produced in successive divisions

In addition to giving rise to pPGCs, 4d (M) blastomere is the origin of the trunk mesoderm in *P. dumerilii* (*Ackermann et al., 2005*). The trunk somatic mesoderm of *P. dumerili* larvae is organized in three largely similar segmental blocks (*Brunet et al., 2015*). Each mesodermal segment is composed of identical left and right halves called mesodermal hemisegments. In clitellate annelids, ML and MR divide asymmetrically giving rise to an anterior to posterior linear series of smaller segmental founder cells called 'primary blast cells' (PBCs). There are as many PBCs on each side as there are segments. Each PBC generates a metamerically iterated clonal mesodermal domain distributed over three consecutive hemisegments. However, each PBC makes 'one hemisegment-worth' of mesoderm (*Gline et al., 2011*; *Weisblat and Kuo, 2014*). This stereotypic, anterior to posterior-oriented, asymmetric division process is termed teloblastic growth. In *P. dumerilii*, to determine whether the 4d lineage (after the divisions that generate pPGCs) makes the mesodermal hemisegments in the

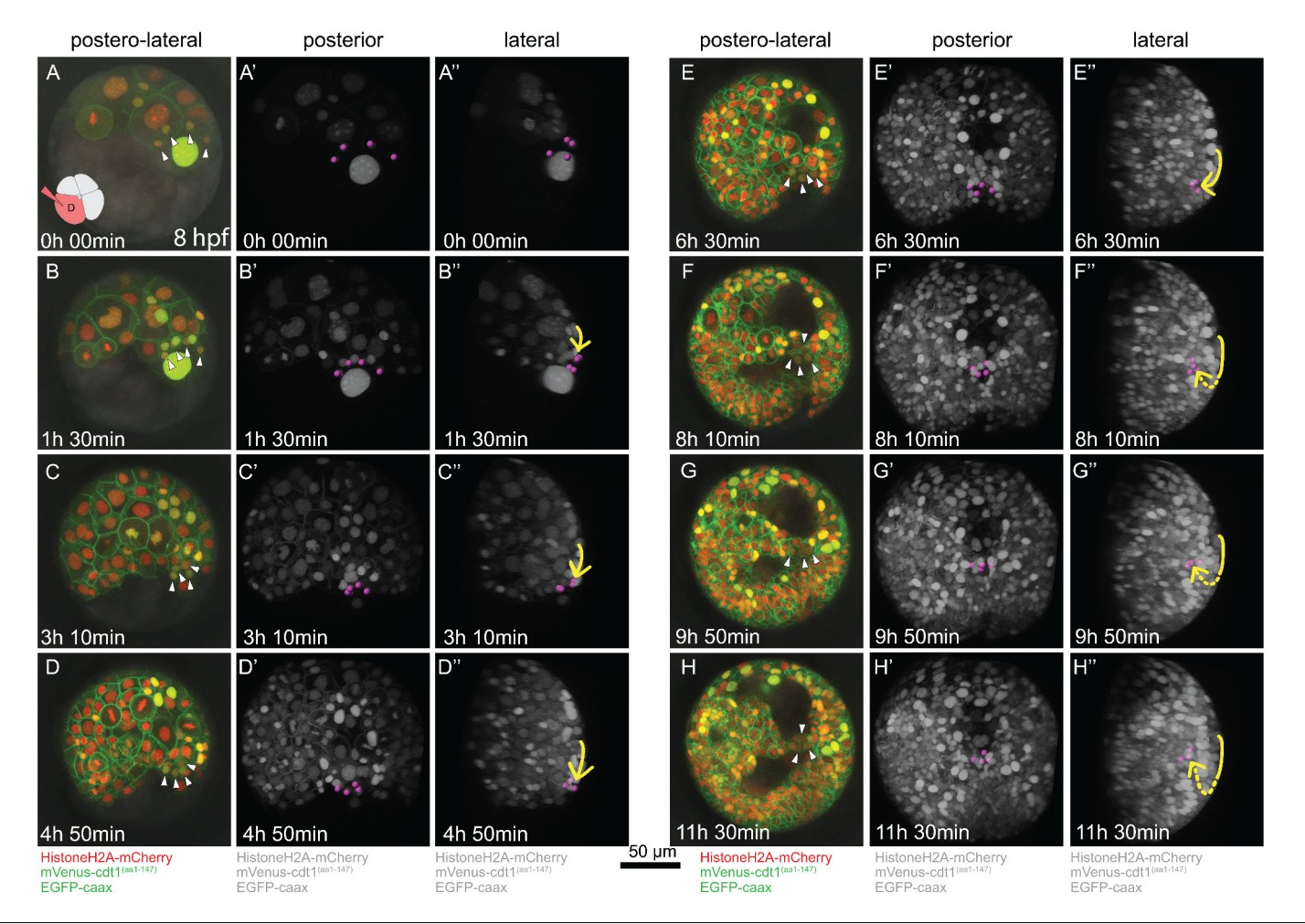

**Figure 3.** Time-lapse series of pPGC migration. Embryo (Sample A) was injected with Pdu-FUCCI and EGFP-caax in the D quadrant and live-imaged with high time resolution (every 10 min). Time-lapse imaging started at 8 hpf and continued for 12 hr. Selected snapshots (elapsed time indicated on the lower-left corner) shown in the figure. (A–H) Z-projections from subsets of stacks showing the sample from a postero-lateral orientation, dorsal up, ventral down. pPGCs are indicated with arrowheads. (A'–H') Same dataset analyzed in Imaris as Z-projection of full stacks, and rotated to a perfect posterior view in the software's 3D viewer. All fluorescent signals are shown in grey color (set artificially in Imaris) and pPGCs (pink spots) were traced using the spots feature (*Figure 3—video 3*). (A''–H'') Lateral view of the same Imaris dataset, posterior to the right, dorsal up (*Figure 3—video 4*). Yellow arrow shows the approximate path of pPGCs as they migrate. Note that they travel on the surface (until E''), then start assuming an internal position (indicated with the dashed yellow line) with the onset of epiboly. For the original video showing all time points and full-projection of stacks see *Figure 3—video 1*. For close-ups see *Figure 3—figure supplement 1*. (In the rest of the manuscript, this sample is analyzed for other developmental processes and is referred to as Sample A. Information from additional samples (Samples B and C) is provided in figure supplements.).
DOI: https://doi.org/10.7554/eLife.30463.010

The following video and figure supplements are available for figure 3:

**Figure supplement 1.** Close-ups of time-lapse video of pPGCs.
DOI: https://doi.org/10.7554/eLife.30463.011

**Figure supplement 2.** The first two divisions of ML and MR.
DOI: https://doi.org/10.7554/eLife.30463.012

**Figure 3—video 1.** Original 3D-projected time-lapse dataset with annotations (for Sample A).
DOI: https://doi.org/10.7554/eLife.30463.013

**Figure 3—video 2.** Video of stack showing the same individual from *Figure 3—video 1* imaged next day.
DOI: https://doi.org/10.7554/eLife.30463.014

**Figure 3—video 3.** pPGC migration traced in Imaris, posterior view.
DOI: https://doi.org/10.7554/eLife.30463.015

**Figure 3—video 4.** pPGC migration traced in Imaris, lateral view

*Figure 3 continued on next page*

*Figure 3 continued*

DOI: https://doi.org/10.7554/eLife.30463.016

larva via the teloblastic process as in the clitellates, we followed divisions of this lineage and analyzed the origin of mesodermal cell clusters (4d lineage is summarized in *Figure 4A*). If mesoderm formation in *P. dumerilii* follows a teloblastic process, a key expectation is the presence of clonally homogeneous mesodermal regions, each originating from one specific primary blast cell as in the clitellate annelids. To do this, we manually traced cell lineages originating from 4d blastomere in live imaging datasets using Imaris software (for details, see Materials and methods). Here we only report data for the left mesoblast (ML) (Sample A), but confirmed in other samples (Samples B and C) that MR behaves like ML (*Figure 5—figure supplement 1*, *Figure 5—video 1*, *Figure 5—video 2*).

Imaris creates a temporal series of fully re-orientable 3D reconstructions of the embryo, allowing the manual tracking of cell divisions. Spots corresponding to the nucleus of each cell are generated and cell lineages can be connected via these spots across time (*Figure 4—video 1*). In addition to nuclear divisions, the cell cycle reporter helped us predict when cell division was approaching, and the cell membrane rounding (visualized via the EGFP-caax construct) helped us detect without ambiguity each mitosis and the resulting daughter cells. Once a cell lineage tree is available, subsets of lineages can be highlighted to visualize where the progeny is located in the 3D reconstruction. Using this feature, we highlighted the progeny of each blast cell that splits from ML (*Figure 4*). We looked at the origin and clonality of cells that formed each segmental mesoderm. Here, we adopt a similar three-character nomenclature (rank, fate, and side) (*Fischer and Arendt, 2013*; *Lyons et al., 2012*). For example, 5ml is the small daughter, of the fifth asymmetric division of the left M teloblast, whereas its sister cell, 5ML is the self-renewed M teloblast.

*P. dumerilii* has three true larval segments with bundles of chaetae on each side and one cryptic anterior segment (also called 'segment 0' [*Saudemont et al., 2008*], or 'segment I' [*Steinmetz et al., 2011*]). The cryptic segment does not bear chaetae and is incorporated into the head of the larva (*Brunet et al., 2015*; *Fischer et al., 2010*; *Steinmetz et al., 2011*). We adopt the following order from anterior to posterior progression: cryptic segment, segments 1, 2, and 3. The numbered segments are already clearly identifiable in the final stacks of our videos corresponding to mid-trochophore stage, because two ectodermal pockets corresponding to chaetal sacs are starting to form in each hemisegment, and mesodermal tissues that will make the segmental musculature are starting to surround them (*Figure 3—video 1*, last frame). We identified four clear mesodermal sublineages that make distinct clonal domains of segmental mesodermal precursor cells arranged in bilateral anterior to posterior series (*Figure 4D–G*, *Figure 4—figure supplement 1*). We found that, progeny of each 'm' blast cell from 6ml to 8ml contributes mostly to a specific larval segment: 6ml contributes to the left larval hemisegment 1 (*Figure 4E–E''''*, *Figure 4—video 8*, *Figure 4—video 9*); 7ml to the left larval hemisegment 2 (*Figure 4F–F''''*, *Figure 4—video 10*, *Figure 4—video 11*); and 8ml to the left larval hemisegment 3 (*Figure 4G–G''''*, *Figure 4—video 12*, *Figure 4—video 13*). These clonal blocks were easily discernible at earlier stages of development as well (*Figure 4—figure supplement 1*). In addition, 5ml proliferates to give a mesodermal block located just anterior to the three true segmental blocks originating from 6ml-8ml and will possibly give rise to the mesoderm of the left cryptic larval hemisegment (*Figure 4D–D''''*, *Figure 4—video 6*, *Figure 4—video 7*). These lineages are summarized in *Figure 4—video 16*.

To confirm that the clonal blocks of cells generated by each blast cell corresponded to segmental anlagen, we bring forward additional observations. First, we further analyzed the dorso-ventral position of the clonal mesodermal clusters at the final time point (t = 72, or about 37.5 hpf of development (*Table 1*) of the time lapse dataset. At this time point, chaetal sacs (ectodermal structures used as anatomical markers of larval segment formation) had already started forming with a few elongating bristles evident, and each clonal mesoderm cluster from a single PBC overlapped with a pair of chaetal sacs present in each hemisegment (*Figure 4—figure supplement 2A–B'*,). Then we analyzed the cross-sections of the 3D stacks with the Imaris lineage spots. We found that spots marking the mesodermal lineage lie just beneath each pair of chaetal sacs within a larval hemisegment (*Figure 4—figure supplement 2C–D''*), confirming that these were mesodermal cells forming the layer beneath the elongating chaetal sacs at this stage. Second, we analyzed another time-lapse movie

showing the development of the trochophore larva up to a stage where segmentation becomes thoroughly apparent (26–54 hpf) (*Figure 4—figure supplement 3*, *Figure 4—video 17*). Mesodermal blocks were initially beneath the elongating and deepening chaetal sacs (*Figure 4—figure supplement 3A–B*, *Figure 4—video 17*), but they eventually formed mesodermal pockets surrounding the chaetal sacs like sheaths (*Figure 4—figure supplement 3C–D*, *Figure 4—video 17*). Interestingly, before the chaetal sacs started to elongate and sink into the mesodermal layer, clear mesodermal segment boundaries were briefly visible, in the form of dorso-ventrally stretched cells (*Figure 4—figure supplement 3C*, arrows). In other models, these boundaries have been shown to correspond to acto-myosin cables that act as active compartmental limits preventing cell mixing (*Monier et al., 2011*). Furthermore, in *P. dumerilii*, previous works have shown that segmental anlagen become distinguishable at the transcriptional level already by 34 hpf, evident by the striped pattern of gene expression in the pre-segmental mesoderm and ectoderm (*Saudemont et al., 2008*; *Steinmetz et al., 2011*). Thus, even though segments are not anatomically distinct yet, at the molecular level mesodermal clusters start to show evidence of segmentation. Together, these observations support early determination of segmental anlage and suggest that there is minimal (if any) mixing of mesodermal cells across segments at the stages we carried out the mesodermal cell lineage analysis and after.

We also looked into the fate of blast cells segregated prior to these segmental lineages. 1ml and 2ml (1mr and 2mr for the right side) corresponded to the bilateral pairs of pPGCs as described above (*Figure 4A–B*). These cells did not divide further and stayed in the immediate vicinity of the M teloblastic cells near the vegetal pole of the embryo. 3ml and 4ml blast cells, unlike 5ml to 8ml, underwent a limited number of symmetric mitoses. These small 3ml and 4ml lineages drifted toward the animal pole. Due to the orientation of the sample during imaging, it was not possible to trace the lineages 3ml and 4ml confidently after a certain time point. Our confocal stacks cover only about 60 μm thickness (the embryos are 160 μm) from the posterior end, and the progeny of 3ml and 4ml move out of the imaged area. However, the limited information we obtained suggests they may be giving rise to anterior-most non-segmental mesodermal tissues (*Figure 4—videos 2–5*). Overall, the data support that ML shows teloblastic behavior and the progeny of mesodermal primary blast cells contribute to distinct larval segments similar to what is observed in clitellate annelids.

## Mesodermal posterior growth zone originates from 8ML and 8MR

At the eighth division of ML and MR, cells 8ML/R and 8ml/r are generated. As described above, 8ml/r are the founder cells of the third segmental mesoderm (*Figure 4G*, *Figure 4—figure supplement 1*, *Figure 4—video 16*). We next investigated the progeny of 8ML/MR (*Figure 5*). We compared three separate individuals (Samples A, B, C) for the 8ML and 8MR divisions and found them to be similar in all samples (*Figure 5—figure supplement 1*, *Figure 5—video 1*, *Figure 5—video 2*). At the time they were born, 8ML/MR were positioned immediately next to the pPGCs (*Figure 6U–U'*; *Figure 4—figure supplement 1*; *Figure 4—video 14*, *Figure 4—video 15*). By mid-trochophore stage (around time points 55–60), all individuals had three cells on the left and right sides near pPGCs (*Figure 5—figure supplement 1A–C'*). Two of these cells divided eventually (around the same time in the left and right sides, but not perfectly simultaneously) (*Figure 5A'–A'''*; *Figure 5—figure supplement 1A''-C''*) expanding the progeny of 8ML and 8MR to 5 cells on each side (10 total). Thus, by the last time point of our datasets (corresponding to the end of mid-trochophore stage), 8ML and 8MR had given rise to five cells each that were surrounding the four pPGCs and formed a posterior 'bridge' between the right and left mesodermal bands. They were positioned slightly posterior to the pPGC cluster (*Figure 5B–B''*), in the region which has been previously described as the mesodermal posterior growth zone (*Rebscher et al., 2007*). We thus suggest that these cells represent the founding lineage ring of mesodermal stem cells that will be active later in juvenile development in the segment addition zone (*Gazave et al., 2013*).

## Both the asymmetry and orientation of embryonic mesoteloblast divisions gradually change

We analyzed the 4d lineage divisions at single-cell resolution for size asymmetry. The division asymmetry (based on cell size) for 4d's two daughter cells ML and MR has been previously described using bright-field microscopy only until the seventh division (7ML/R and 7ml/r) conclusively

**Table 1.** Calculated developmental time and corresponding stage for each time point at imaging temperatures.

For the video (Sample A) from which we show most of the lineage tracing data in this manuscript, we calculated what each time point roughly corresponds to in the normal staging by *Fischer et al., 2010*. This calculation was done based on the starting stage, end stage, and how long this normally takes under 18°C conditions. For example, *Figure 3—video 1* (used in *Figures 3*, *4*, *5* and *6*) starts at eight hpf (stereoblastula stage), and in the final time point the sample appears towards the end of mid trochophore stage. Even though elapsed time is 12 hr of imaging (72 time points), under 18°C incubation conditions, about 35–40 hr is required to reach this stage. Thus, we calculated each time point in the live-image dataset corresponding roughly to: T = 8 hr + (25mins x tp), where 'tp' is time point, and T is the calculated developmental time. According to this, the end calculated developmental time for *Figure 3—video 1* is: T = 8 hr + (25 mins x 72)=37 hr 35 min.

| Time point (tp) | Elapsed imaging time (hr: min:s) | Calculated developmental time (T) at 18°C (hr:min:s) | Corresponding developmental stage at 18°C (*Fischer et al., 2010*) |
|---|---|---|---|
| 1 | 0:00:00 | 8:00:00 | Stereoblastula |
| 3 | 0:20:00 | 8:50:00 | Stereoblastula |
| 5 | 0:40:00 | 9:40:00 | Stereoblastula |
| 7 | 1:00:00 | 10:30:00 | Stereoblastula |
| 9 | 1:20:00 | 11:20:00 | Stereoblastula |
| 11 | 1:40:00 | 12:10:00 | Stereoblastula |
| 13 | 2:00:00 | 13:00:00 | Protrochophore |
| 15 | 2:20:00 | 13:50:00 | Protrochophore |
| 17 | 2:40:00 | 14:40:00 | Protrochophore |
| 19 | 3:00:00 | 15:30:00 | Protrochophore |
| 21 | 3:20:00 | 16:20:00 | Protrochophore |
| 23 | 3:40:00 | 17:10:00 | Protrochophore |
| 25 | 4:00:00 | 18:00:00 | Protrochophore |
| 27 | 4:20:00 | 18:50:00 | Protrochophore |
| 29 | 4:40:00 | 19:40:00 | Protrochophore |
| 31 | 5:00:00 | 20:30:00 | Protrochophore |
| 33 | 5:20:00 | 21:20:00 | Protrochophore |
| 35 | 5:40:00 | 22:10:00 | Protrochophore |
| 37 | 6:00:00 | 23:00:00 | Protrochophore |
| 39 | 6:20:00 | 23:50:00 | Protrochophore |
| 41 | 6:40:00 | 24:40:00 | Early trochophore |
| 43 | 7:00:00 | 25:30:00 | Early trochophore |
| 45 | 7:20:00 | 26:20:00 | Mid-trochophore |
| 47 | 7:40:00 | 27:10:00 | Mid-trochophore |
| 49 | 8:00:00 | 28:00:00 | Mid-trochophore |
| 51 | 8:20:00 | 28:50:00 | Mid-trochophore |
| 53 | 8:40:00 | 29:40:00 | Mid-trochophore |
| 55 | 9:00:00 | 30:30:00 | Mid-trochophore |
| 57 | 9:20:00 | 31:20:00 | Mid-trochophore |
| 59 | 9:40:00 | 32:10:00 | Mid-trochophore |
| 61 | 10:00:00 | 33:00:00 | Mid-trochophore |
| 63 | 10:20:00 | 33:50:00 | Mid-trochophore |
| 65 | 10:40:00 | 34:40:00 | Mid-trochophore |
| 67 | 11:00:00 | 35:30:00 | Mid-trochophore |
| 69 | 11:20:00 | 36:20:00 | Mid-trochophore |
| 71 | 11:40:00 | 37:10:00 | Mid-trochophore |
| 72 | 11:50:00 | 37:35:00 | Mid-trochophore |

DOI: https://doi.org/10.7554/eLife.30463.017

(*Fischer and Arendt, 2013*), and the divisions after this were not possible to observe due to the limitations of this imaging technique. We took advantage of high-resolution confocal imaging with injection of the fluorescent markers Pdu-FUCCI (nuclear) and EGFP-caax (cell membrane, making the cell size very easy to observe). Here, we show the imaging dataset of an embryo (Sample A, same dataset as reported above) injected into its D quadrant and live-imaged from 8 hpf until larval stages when segmental mesoderm morphology started becoming obvious (*Table 1*, *Figure 3—video 1*). In the confocal stacks, we were able to investigate division of 4d lineage further than previously reported, and we analyzed the size asymmetry as well as cell division angle for this lineage.

The first division of 4d is equal, creating two sister mesoblast cells positioned bilaterally symmetrically (ML and MR) (*Figure 6A–B*, *Figure 3—figure supplement 2*). However, most of the rest of the divisions were unequal, as expected of teloblasts. The first two divisions of ML and MR, are the most unequal. They gave rise to large daughter cells (1ML, 1MR, 2ML, 2MR) and very small ones (1ml, 1mr, 2ml, 2mr). 1ml/r and 2ml/r are the pPGCs as explained above (*Figure 4A–C*; *Figure 3—figure supplement 2*). For the next rounds of divisions, we report the data for ML in detail, but we confirmed MR behaves similarly in two more samples investigated (*Figure 5—figure supplement 1*, *Figure 5—video 1*, *Figure 5—video 2*). After the initial divisions that gave rise to the pPGCs, the mesoblast (now named 2ML) continued to divide, budding off small cells: division of 2ML, 3ML, 4ML, 5ML were all unequal (*Figure 6E, G, J and L*, respectively). However, the size difference between the two daughter cells progressively decreased. This seemed to be due to the fact that 3ml and 4ml were small precursors compared to the later segmental precursors, 5ml, 6ml (compare *Figure 6E and G with 6J and L*) but also due to the gradual decrease in size of the remaining M teloblast. Following this tendency, division of 6ML into 7ML and 7ml was roughly equal in size (*Figure 6O*). We observed that 7ML then divided unequally again, giving rise to 8ML and 8ml (*Figure 6S*). But, surprisingly this time, the size difference was inverted: 8ml, precursor of the mesoderm of the third segment was the large cell and 8ML, the remaining teloblast cell, was now very small. The further divisions of this remaining teloblast 8ML are essentially equal and give a cluster of quite small cells. Thus, we confirm that 4d lineage shows asymmetric divisions similar to clitellate mesoteloblasts. However, unlike in the leech, in *P. dumerilii* the mesodermal teloblasts decreased in size very rapidly through this eight-division sequence, and we observed a reversed asymmetry at the end of the sequence. The mesodermal teloblastic lineage (derived from 8ML/MR) that will presumably produce the segments of the juvenile thus starts from a cluster of very small cells.

As observed in previous work (*Fischer and Arendt, 2013*), the orientation of divisions also sequentially shifted in a clock-wise fashion for the left mesoteloblast divisions (*Figure 6E, G, J, L, O and S*, summarized in *Figure 6V*), when looking from the posterior pole. Consequently, whereas the first series of precursors (pPGCs, 3ml, 4ml) were budded off roughly in the direction of the future blastopore, the later segmental precursors (5ml to 8ml) were budded off progressively more towards the anterior of the embryo, also reflecting the gradual change of size of the teloblast cell and its shifting position toward the vegetal pole.

## Divisions generating segmental mesoderm precursors happen in quick succession, then cell cycling slows down significantly

We assessed whether the divisions giving rise to segmental precursors follow some kind of 'segmentation clock', comparable to the way somites are produced at a regular time interval in vertebrates (*Cooke and Zeeman, 1976*). This is not the case in *P. dumerilii* as the time interval increases gradually between divisions. Precisely, at the experimental temperatures 4ML, 5ML, 6ML, 7ML divisions happened in 30, 30, 40, and 50 mins after the preceding division, respectively. Thus, there is no 'segmentation clock' but a 'segmentation oscillator' of decreasing frequency in the embryo.

We next used the live cell cycle reporter Pdu-FUCCI to compare the cell cycling patterns of some of the different lineages that arise from the 4d micromere. To do this, mean fluorescence intensity information was extracted for each spot from the cell lineage datasets using statistics module in Imaris and these were plotted as graphs (*Figure 7A*, *Figure 7—source data 1*). Each spot was manually placed inside each nucleus (*Figure 7A'*), and mean fluorescence intensity of pixels within each spot

was calculated for subsets of lineages (see Materials and methods for details). The measurement reflects the abundance of mVenus-Cdt1(aa1-147) protein, providing quantitation of the cycling of the construct. pPGCs stopped cycling after they were born (*Figures 2* and *3*; *Figure 3—figure supplement 1*) and as expected, a roughly linear plot, without significant valleys and peaks was obtained for these cells (*Figure 7B*). The mesoblast ML, in contrast, had clear cycling with prominent valleys and peaks (*Figure 7B*, *Figure 7—figure supplement 1A*). Next, we selected a sub-lineage within primary blast lineages 7ml, and 8ml, as well as a sub-lineage within 8ML for analyzing cell cycling patterns. In segmental mesoderm lineages within 7ml and 8ml, we selected 'equivalent' sub-lineages based on the location of daughter cells as the primary blast cell divided (see Materials and methods for details of the lineage selection criteria). The comparison of 7ml sub-lineage to 8ml sub-lineage revealed that initially these two are out of phase (valleys and peaks do not overlap) even though cycling periodicity is similar. However, by time point 46 they become synchronized as shown by the overlapping graphs (*Figure 7C*). On the other hand, 8ml and 8ML sub-lineages differ from each other drastically: 8ML has much fewer valleys and peaks (*Figure 7D*, *Figure 7—figure supplement 2A–E*, *Figure 7—figure supplement 1C*), indicating slower cell cycling compared to 8ml (*Figure 7D*, *Figure 7—figure supplement 2F–J*, *Figure 7—figure supplement 1B*). We have also confirmed that the cycling patterns are similar in different samples and in the right mesoblast (*Figure 7—figure supplement 2*). Overall, these data show that live cell cycle reporter Pdu-FUCCI is a useful tool for revealing changing cycling signatures within lineages and across lineages. This tool can be used for similar purposes in other developmental processes and stages.

## Discussion

Here we report techniques for long-term live imaging of *P. dumerilii* embryos and larvae, and demonstrate the feasibility of a FUCCI live-cell cycle reporter in this species. Using these techniques and tools, we reveal previously unknown characteristics of the development of this marine annelid, in particular the fact that the daughters of the blastomere 4d (ML and MR) produce the trunk mesoderm by teloblastic divisions, as in clitellate annelids. Starting with the fifth division, each division of this pair of cells gives rise to the mesoderm of one segment. ML and MR also give rise, by their two first divisions, to the putative Primordial Germ Cells (pPGCs), and starting with the eighth round of division, to the mesodermal posterior growth zone (MPGZ), which is part of the segment addition zone (SAZ). Interestingly, the different cell types that originate from 4d (segmental mesoderm cells, posterior growth zone cells, and germ cells) show distinct cell cycle characteristics.

### Live imaging the 4d (M) lineage in *P. dumerilii* with a live-cell cycle reporter

Long-term imaging at single-cell resolution for lineage analyses has been technically challenging in most spiralians including annelids. The micromere lineages (such as 4d) are small and very difficult to label individually in non-clitellate annelids (i.e. non-clitellate Sedentaria, Errantia, and basal annelids) (*Irvine and Seaver, 2006*), as opposed to the leech and the sludge worm *Tubifex* (both clitellates), which have larger and accessible embryos that enable injecting single blastomeres to follow specific lineages (*Gline et al., 2011*, *Gline et al., 2009*; *Goto et al., 1999a*, *1999b*; *Storey, 1989*; *Weisblat and Shankland, 1985*). Despite the difficulty of injections, cell lineage analyses have been carried out nevertheless. Some of the studies used DiI injections for labeling specific blastomeres and analyzing the domains they gave rise to in fixed samples, in which case there is no continuous observation by time-lapse imaging, thus no resolution at individual cell-level (*Ackermann et al., 2005*; *Irvine and Seaver, 2006*). Other studies used bright-field microscopy for live observations, but only earlier stages of development could be analyzed as cell sizes become too small at later stages to confidently follow without any labeling, and/or the larvae start moving due to ciliary band formation (*Fischer and Arendt, 2013*; *Wilson, 1892* in two other nereidids, *Nereis limbata* and *Platynereis megalops*). As a result, behavior of specific lineages with detailed cell cycling characteristics at single-cell resolution could not be observed, and whether the lineages showed teloblastic activity in non-clitellate annelids could not be directly observed. Taking advantage of injecting mRNAs coding for fluorescently labeled proteins, confocal imaging, and specimen immobilization, we were able to live-image the 4d lineage until late trochophore stages (summarized in *Figure 8*) and analyzed its cell cycling patterns. Overcoming the long-standing challenge

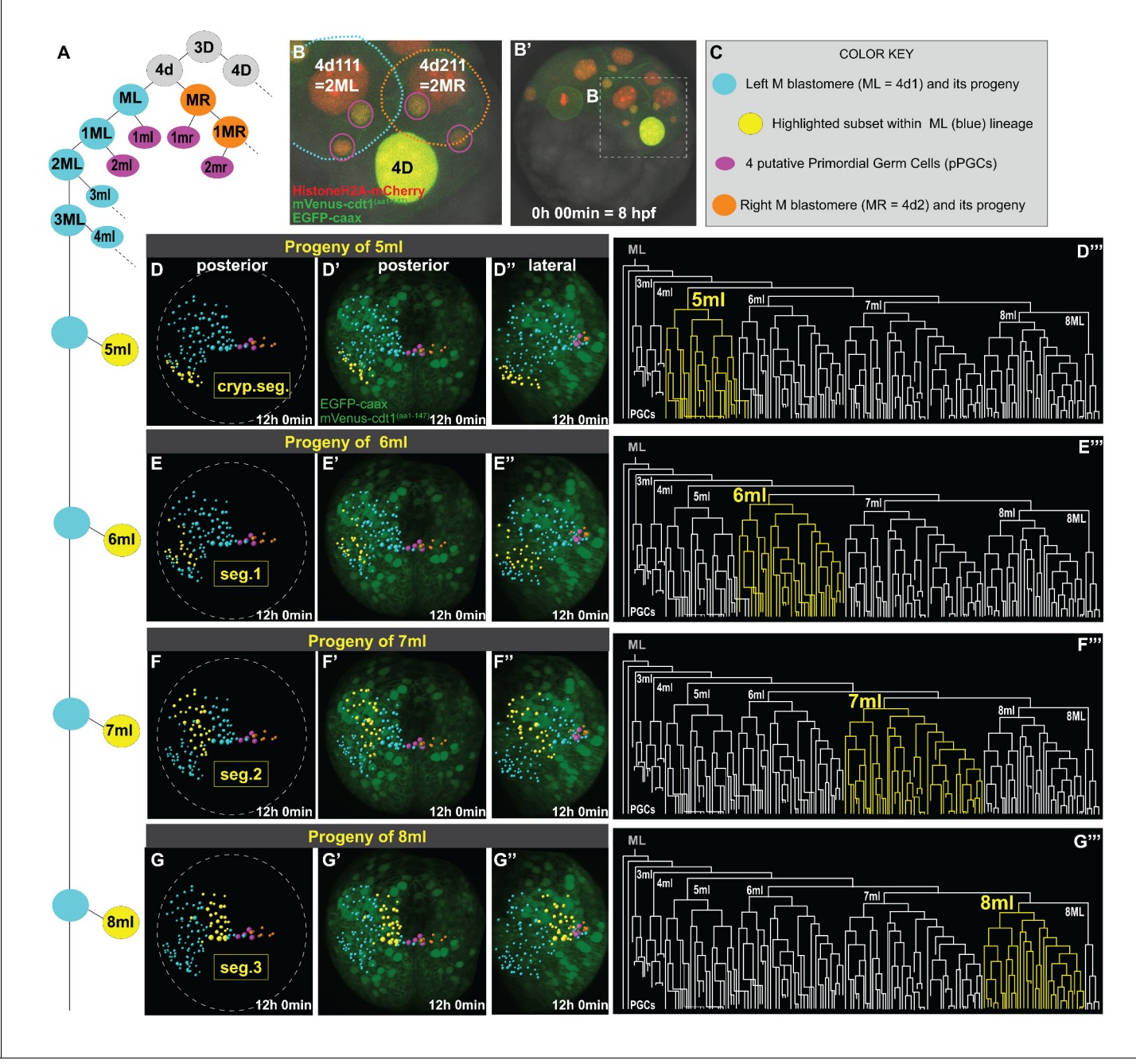

**Figure 4.** Primary blast cells give rise to clonal bocks of mesoderm.  The same sample as shown in *Figure 3* (Sample A) was analyzed for mesodermal lineages in Bitplane Imaris. 4-cell stage embryo injected with Pdu-FUCCI (*HistoneH2A-mCherry, mVenus-cdt1(aa1-147)*) and *EGFP-caax* into D quadrant was live-imaged starting at 8 hpf every 10 min for 12 hr, from a posterior-lateral angle which enabled the tracing and visualization of the left mesoblast (ML = 4d1) lineage. Schema in **A** shows 4d lineage: ML (blue), MR (orange), pPGCs (pink), and highlighted subsets (yellow) are all color-coded (summarized in **C**) throughout this and following figures. Note that only a limited number of cells were traced for MR. Panel B (zoomed-in from **B'**) shows the starting frame of the time-lapse, and the position of ML and MR, with all three fluorescent constructs (see *Figure 3—video 1* for the original time-lapse with annotations, and *Figure 4—video 1* with spots for lineage tracing of ML). In the panels (**D'- G''**), the last time point from this time-lapse (Sample A) is shown with EGFP-caax and mVenus-Cdt1(aa1-147) only (both in green), but HistoneH2A-mCherry (red) was removed for easier observation of the colored spots. 5ml lineage makes the mesoderm of the anterior-most left cryptic hemisegment (**D–D'''**), 6ml makes the 1st larval left hemisegment (**E–E'''**), 7ml the 2nd larval left hemisegment (**F–F'''**), and 8ml the 3rd larval left hemisegment (**G–G'''**). Panels D-G are the same orientation as D'-G' but only showing the spots. **D'–G'** show posterior orientation, and **D''-G''** show lateral orientation. The lineages highlighted in yellow in **D–G''** are also shown in yellow in the lineage trees (**D'''–G'''**). See *Figure 4—video 16* for an animated version of these lineages highlighted. *Figure 4—video 2–13* show the full time-lapse of each lineage from posterior and lateral orientations, as well as 360-degree rotation.

*Figure 4 continued on next page*

*Figure 4 continued*

DOI: https://doi.org/10.7554/eLife.30463.018

The following video and figure supplements are available for figure 4:

**Figure supplement 1.** Progeny of each blast cell at an intermediate time point.

DOI: https://doi.org/10.7554/eLife.30463.019

**Figure supplement 2.** Mesodermal clonal clusters align with chaetal sacs specific to each hemisegment.

DOI: https://doi.org/10.7554/eLife.30463.020

**Figure supplement 3.** Live-imaging of the formation of mesodermal segmental blocks and the corresponding chaetal sacs.

DOI: https://doi.org/10.7554/eLife.30463.021

**Figure 4—video 1.** Time-lapse of the ML lineage, and 360 rotation of the last time point.

DOI: https://doi.org/10.7554/eLife.30463.022

**Figure 4—video 2.** 3ml, posterior view.

DOI: https://doi.org/10.7554/eLife.30463.023

**Figure 4—video 3.** 3ml, lateral view.

DOI: https://doi.org/10.7554/eLife.30463.024

**Figure 4—video 4.** 4ml, posterior view.

DOI: https://doi.org/10.7554/eLife.30463.025

**Figure 4—video 5.** 4ml, lateral view.

DOI: https://doi.org/10.7554/eLife.30463.026

**Figure 4—video 6.** 5ml, posterior view.

DOI: https://doi.org/10.7554/eLife.30463.027

**Figure 4—video 7.** 5ml, lateral view.

DOI: https://doi.org/10.7554/eLife.30463.028

**Figure 4—video 8.** 6ml, posterior view.

DOI: https://doi.org/10.7554/eLife.30463.029

**Figure 4—video 9.** 6ml, lateral view.

DOI: https://doi.org/10.7554/eLife.30463.030

**Figure 4—video 10.** 7ml, posterior view.

DOI: https://doi.org/10.7554/eLife.30463.031

**Figure 4—video 11.** 7ml, lateral view.

DOI: https://doi.org/10.7554/eLife.30463.032

**Figure 4—video 12.** 8ml, posterior view.

DOI: https://doi.org/10.7554/eLife.30463.033

**Figure 4—video 13.** 8ml, lateral view.

DOI: https://doi.org/10.7554/eLife.30463.034

**Figure 4—video 14.** 8ML, posterior view.

DOI: https://doi.org/10.7554/eLife.30463.035

**Figure 4—video 15.** 8ML, lateral view.

DOI: https://doi.org/10.7554/eLife.30463.036

**Figure 4—video 16.** Animation of the segmental mesoderm clonal regions.

DOI: https://doi.org/10.7554/eLife.30463.037

**Figure 4—video 17.** Segment formation in the *P. dumerilii* larva.

DOI: https://doi.org/10.7554/eLife.30463.038

of immobilization of the trochophore larvae was particularly significant, because the larvae start spinning from 15 hpf and later actively swim via several ciliary bands. Thus, live imaging for extended time periods to follow cell lineages has not been possible. Our DMSO treatment method (Materials and methods) for cilia elimination can be potentially applied to other marine ciliated larvae as well as freshwater ciliates (Balavoine, Özpolat, Duharcourt, unpublished results).

We generated a live-cell cycle reporter in *P. dumerilii*, using the gene *Pdu-cdt1* (*Figure 1*). Cdt1 cell cycle component in mammal and zebrafish FUCCI (fluorescent ubiquitination-based cell cycle indicator) is restricted to the G1 phase (*Sakaue-Sawano et al., 2008*; *Sugiyama et al., 2009*). Our reporter mVenus-Cdt1(aa1-147) is not specific to G1, but it was sufficient to observe cell cycling patterns, therefore is a useful tool to study cell lineages and cell cycle dynamics during development of *P. dumerilii.* To our knowledge, this is the first live-cell cycle reporter available in a spiralian model

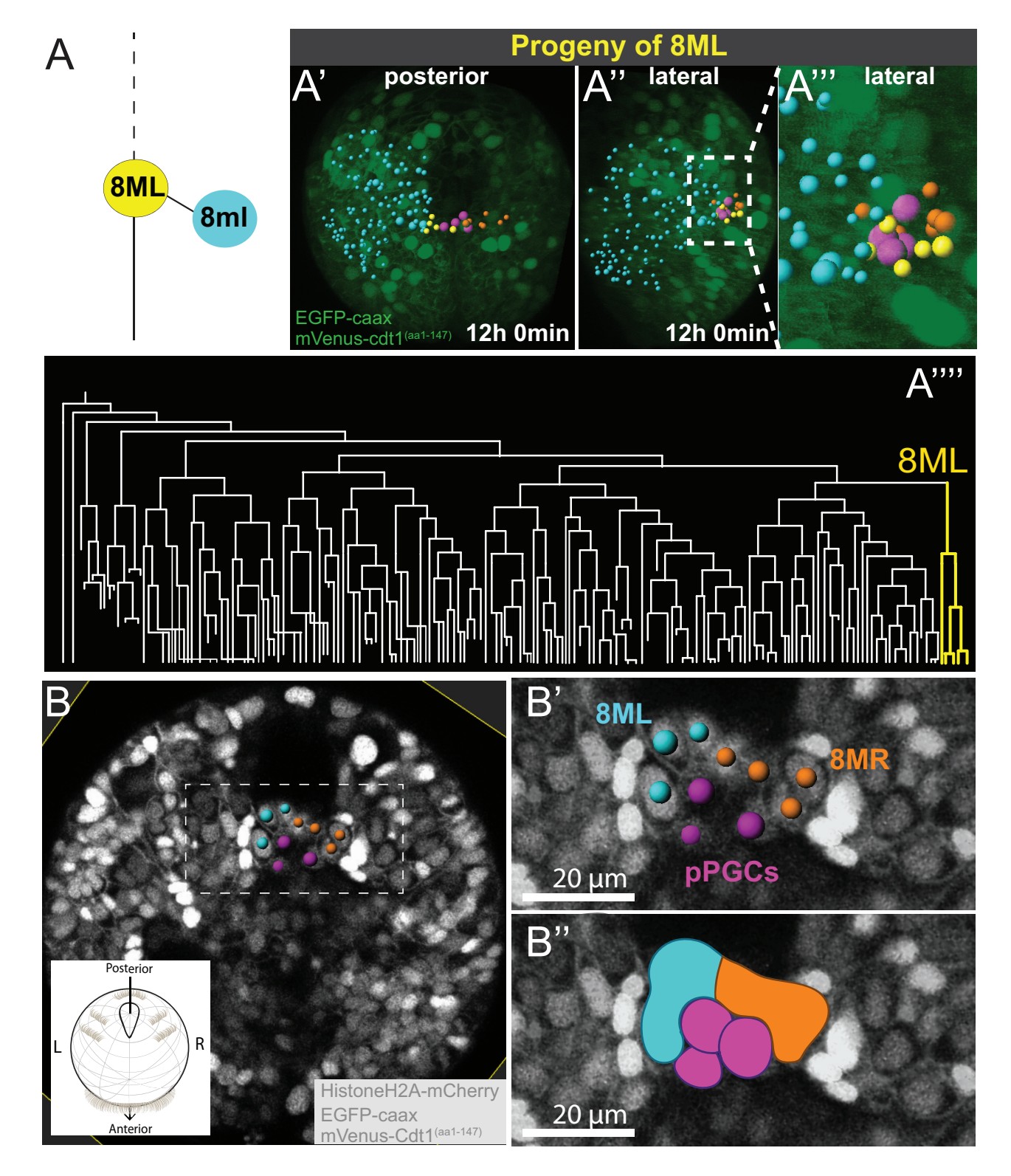

**Figure 5.** 8ML/R give rise to the mesodermal posterior growth zone. Continuation of lineage analysis (in *Figure 4*) of Sample A shows that (**A**) the 8ML cell created during the 8th division gives rise to a few cells near the pPGCs (**A'–A''**), where the mesodermal posterior growth zone is located. The 8ML lineage is highlighted in the tree (**A'''**). Another sample (Sample B) with different orientation is shown in **B-B''**. A subset of 3D-projected focal planes show how the 8ML/8MR lineages are located immediately posterior to the pPGCs (**B'**, **B''**). The same color coding as in *Figure 4* is used: Blue is ML

*Figure 5 continued on next page*

*Figure 5 continued*

lineage, orange is MR, pink is pPGCs, and yellow 8ML. For a comparison of three different samples with 8ML/MR lineages traced see *Figure 5—figure supplement 1*. See *Figure 4—video 14* and *Figure 4—video 15* for full time-lapse with spots showing lineages, from posterior and lateral orientations, as well as 360° rotation (*Figure 4—video 14*). All signals from fluorescent constructs is shown in green (EGFP-caax and mVenus-Cdt1(aa1-147), in A'-A''') or in grayscale (Histone2A-mCherry, EGFP-caax, and mVenus-Cdt1(aa1-147)) to highlight general morphology.
DOI: https://doi.org/10.7554/eLife.30463.039

The following video and figure supplement are available for figure 5:

**Figure supplement 1.** Cell lineages from additional samples compared with the sample shown in the main text.
DOI: https://doi.org/10.7554/eLife.30463.040
**Figure 5—video 1.** Time-lapse of Sample B.
DOI: https://doi.org/10.7554/eLife.30463.041
**Figure 5—video 2.** Time-lapse of Sample C.
DOI: https://doi.org/10.7554/eLife.30463.042

system. Studies are in progress to develop reporters for other phases of cell cycle in *P. dumerilii*, as well as transgenic lines containing one or more of the live cell cycle constructs.

## Cell cycle patterns of the 4d lineage

4d is a conserved blastomere across many spiralian organisms with diverse body plans (*Lambert, 2008*; *Lyons et al., 2012*). However, only limited data are available on detailed 4d lineage/fate maps, cell cycle characteristics, and how cell cycle regulation is related to different cell types '4d program' generates (*Bissen, 1997*, *1995*; *Bissen and Weisblat, 1989*; *Chen and Bissen, 1997*). In *P. dumerilii*, 4d generates differentially fated cell populations, which then display different cycling characteristics. pPGCs (the first lineages to segregate from 4d) are mitotically quiescent and presumably do not divide until later in development. However, lineages that arise right after pPGCs, the precursors of mesoderm, show diverse cell cycling patterns (*Figure 7*). For example, we found early cycles of ML/R to be very short in *P. dumerilii* (*Figure 7—figure supplement 1A*). Then cell cycle becomes slightly longer within mesodermal lineages with relatively extended G2 phase. These findings in *P. dumerilii* are reminiscent of the leech, in which, early ML/R divisions have short cell cycle durations due to absence of G1 and a very short G2 phase. Later cycles in the leech teloblasts become longer due to an increase in the G2 phase length, and appearance of G1 phase (*Bissen and Weisblat, 1989*).

In stark contrast to the fast cycling lineages of the trunk mesoderm (3ml/r-8ml/r), precursors of the MPGZ (8ML/R) are a slow cycling lineage which has relatively short G1 and long G2 phases (*Figure 7—figure supplement 1C*). These cells resemble stem cells in other organisms (such as hydra, mammals, *Drosophila*, and *C. elegans*) because of their slow cycling as well as having short G1 and longer G2 phase (*Ables and Drummond-Barbosa, 2013*; *Buzgariu et al., 2014*; *Hsu et al., 2008*; *Roccio et al., 2013*; *Seidel and Kimble, 2015*). How will the cycling characteristics of the growth zone cells change with slow or fast growth (for example, based on nutrient availability), or in response to injury (for example, during segment addition in the regenerated tail, which grows faster) remains to be determined.

## The first lineage segregated from 4d: putative primordial germ cells

In organisms where germline specification takes place early during development, the PGCs are segregated before gastrulation (*Extavour and Akam, 2003*). This also appears to be the case in *P. dumerilii*. We confirmed the earlier observations (*Fischer and Arendt, 2013*; *Rebscher et al., 2012*, *Rebscher et al., 2007*) that suggested that four putative PGCs (pPGCs; 1ml/r and 2ml/r cells) arise from the first two divisions of 4d's daughters, ML/R. Our and previous observations suggest these cells make only pPGCs (*Ackermann et al., 2005*; *Rebscher et al., 2012*; *Rebscher et al., 2007*), thus pPGCs are made separately from the segmental mesoderm lineages which arise during the subsequent divisions. Here, we refer to these cells as 'putative' PGCs because a continuous lineage tracing (or genetic lineage tracing) by following these progenitors into adulthood when they would form the gametes has not been carried out. However, these cells fit the PGC criteria used in other model systems: they express germline/multipotency program (GMP) genes (*Juliano et al., 2010*) and Vasa protein, at day 4 of development they migrate anteriorly (*Rebscher et al., 2012*; *Rebscher et al.,*

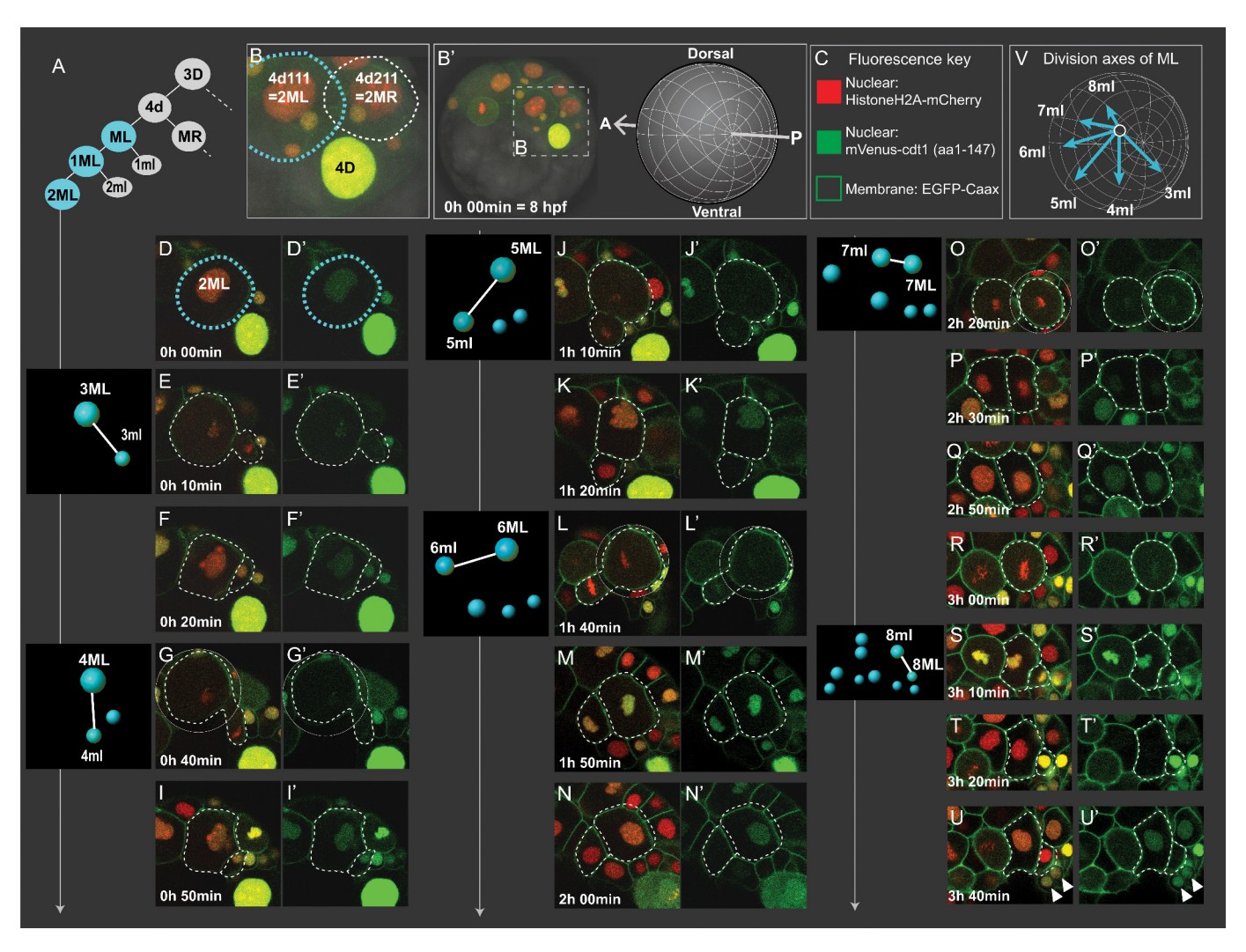

**Figure 6.** Cell divisions of 4d lineage giving rise to the mesodermal primary blast cells. Same dataset shown in *Figures 4* and *5* (Sample A) was analyzed for cell division size asymmetry and orientation. In (**A**), schema shows the 4d lineage divisions, ML in blue. The spot diagrams next to the images show the division by which each primary blast cell is created. The direction of division shown in spot diagrams represents the actual division axis. (**B**) is a zoomed-in panel from (**B'**) showing the mesoblasts after two rounds of division (2ML = 4d111, and 2MR = 4d211). The orientation of imaging is depicted in (**B'**). In (**C**), fluorescence key shows the color and location of each construct. Throughout **D-U** (merged channels) and **D'-U'** (green channel only) snapshots of ML's divisions from the live imaging dataset are shown, outlined with dashed lines. Note that the initial divisions are highly asymmetric in size (**E, G, J, L, S**) while the division that creates 7ml and 7ML appears symmetric (**O**). In **V**, the division axes for all primary blast cells are shown together. See *Figure 3—video 1* for the time-lapse dataset, the snapshots are derived from (Sample A). Arrowheads in (**U, U'**): pPGCs. Time is shown as elapsed imaging time (*Table 1*).

DOI: https://doi.org/10.7554/eLife.30463.043

*2007*), they are mitotically quiescent (*Figures 2* and *3*) but after migration they start proliferating to make future gametes (*Rebscher, 2014*). However, considering that some annelids can regenerate their germline (*Herlant-Meewis, 1946*; *Özpolat et al., 2016*; *Özpolat et al., 2016*; *Stéphan-Dubois, 1964*), these PGCs set aside early in development in *P. dumerilii* may not be the only source of the germ cells in this species. In particular, the ectodermal and mesodermal cells that constitute the SAZ of the juvenile worm persistently express the GMP genes such as *vasa*, *piwi* or *nanos* (*Gazave et al., 2013*; *Rebscher et al., 2007*). It is currently not known whether the GMP is shared between these apparently different sets of cells because the program is crucial for multipotency, or whether the GMP expression in SAZ, and growth zone cells in general, reflects a persistent ability of

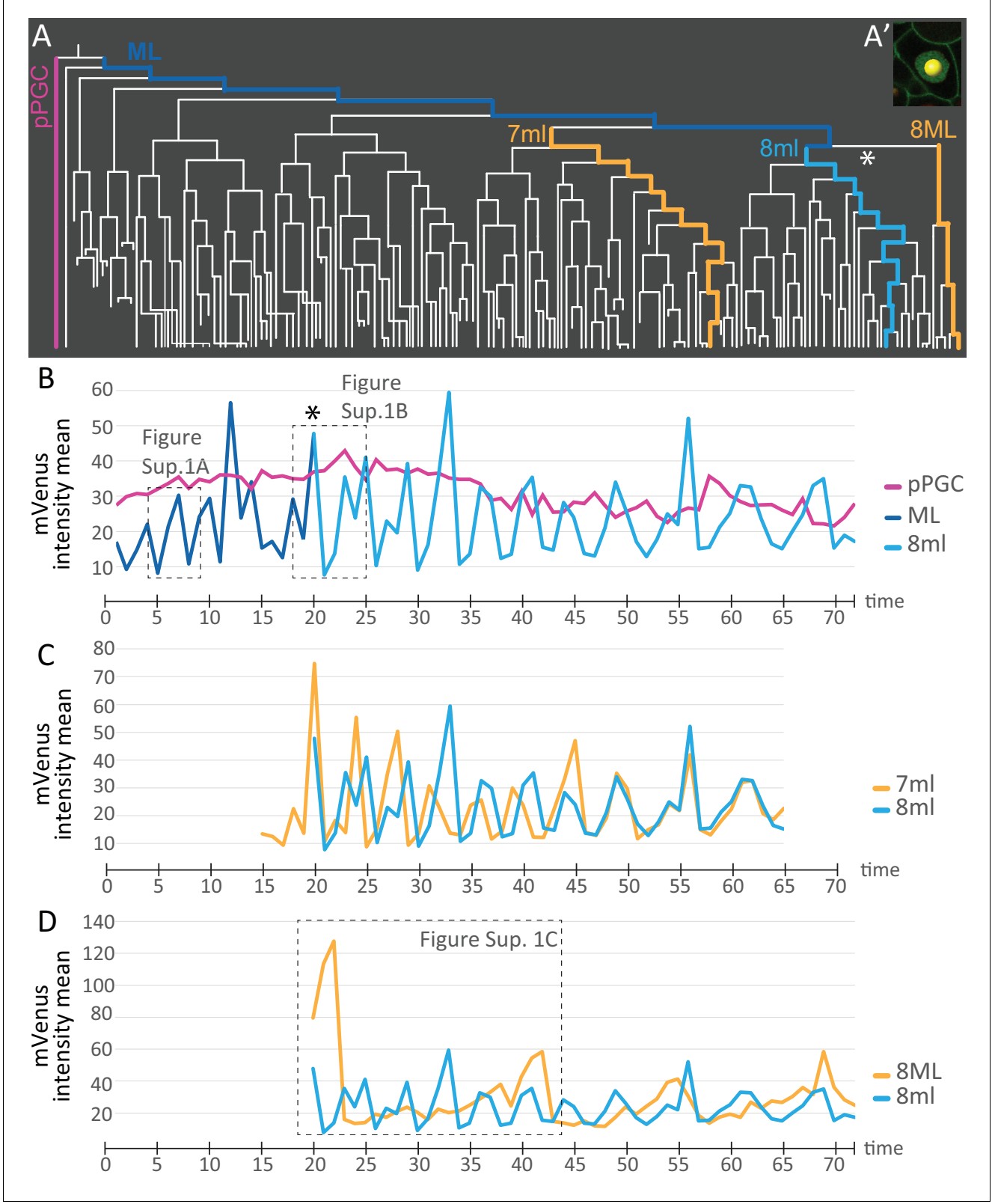

**Figure 7.** Comparison of mVenus-Cdt1(aa1-147) intensity across different cell lineages. (**A**) The lineages selected for cell cycle signature analysis are indicated in color (matching the colors used in the subsequent graphs). (**A'**) Example of a spot placed in the middle of the nuclear signal (only one focal plane from the stack shown here), and used for obtaining 'mVenus fluorescence intensity mean' measurements for each nucleus per time point in Imaris. (**B**) Comparison of mVenus intensity across 3 cell lineages. 8ml (light blue) is plotted as a continuation of the mesoblast ML (dark blue). Star

*Figure 7 continued on next page*

**Figure 7 continued**

indicates the time point 7ML divides into 8ml and 8ML in the lineage tree (**A**) and the cell cycle graph (**B**). (**C**) Two lineages from 7ml and 8ml show similar cycling patterns. Despite being slightly out of phase at the beginning, they become synchronized around time point 46. (**D**) Same 8ml lineage is compared this time with a lineage from 8ML, which is the lineage that gives rise to the mesodermal posterior growth zone cells, and is much slower in its cycling. In B and D, the dashed boxes show the excerpts from the graphs analyzed in *Figure 7—figure supplement 1*. Time axis shows time points (*Table 1*). Fluorescence measurement is in arbitrary units (*Figure 7—source data 1*).

DOI: https://doi.org/10.7554/eLife.30463.044

The following source data and figure supplements are available for figure 7:

**Source data 1.**
DOI: https://doi.org/10.7554/eLife.30463.047
**Figure supplement 1.** Details from the cell cycle graphs.
DOI: https://doi.org/10.7554/eLife.30463.045
**Figure supplement 2.** Cell cycle graphs compared across different live-imaged samples.
DOI: https://doi.org/10.7554/eLife.30463.046

these cells to produce germline cells. A similar situation has been described in planarians, where both the neoblast cells (the planarian stem cells) and the germ cells express *piwi* (*Handberg-Thorsager, 2008*; *Nakagawa et al., 2012*). In planarians, the neoblasts give rise to the germ cells (*Morgan, 1901*; *Newmark et al., 2008*; *Solana, 2013*). Only further investigation involving genetic cell

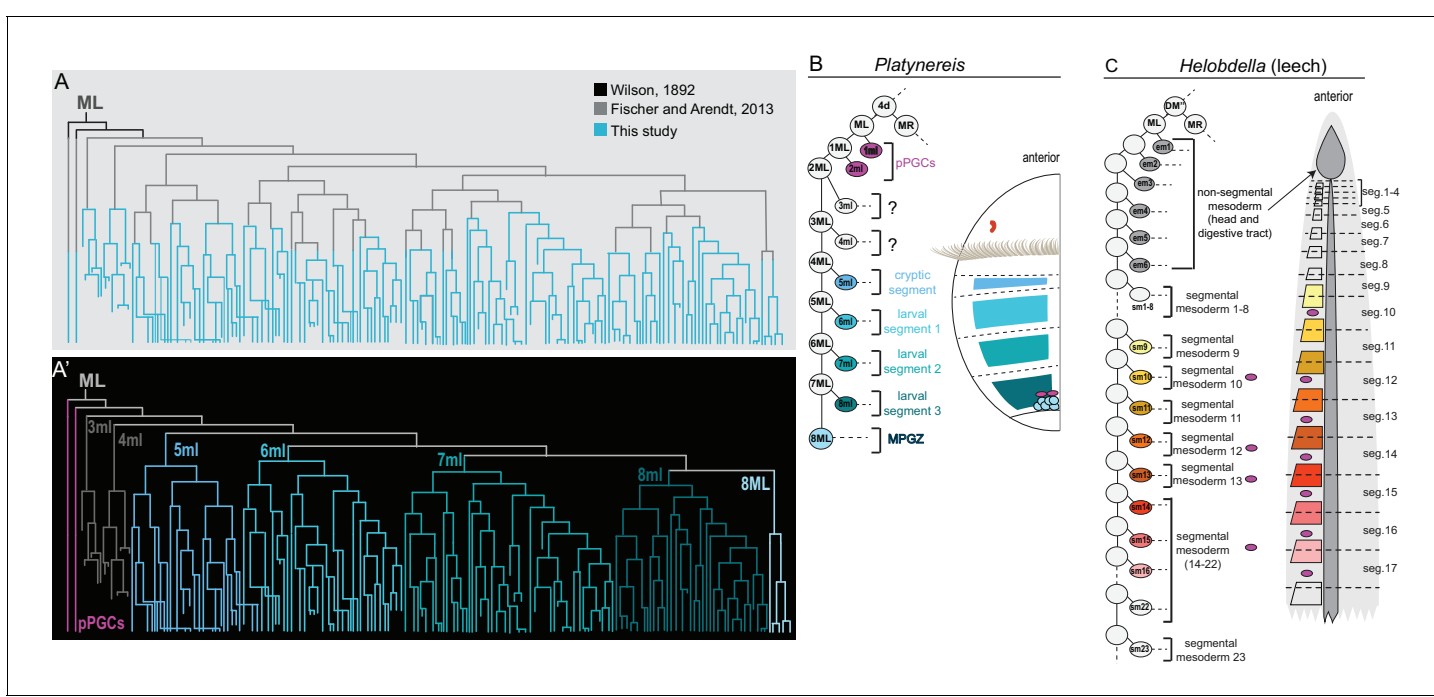

**Figure 8.** ML lineage tree indicating known and novel sub-lineages in *P. dumerilii*, and comparison of *P. dumerilii* with the leech. Lineage tree of cells that arise from the left mesoblast (ML) traced in Imaris is plotted in **A**. Colors show the extent of lineage analyses accomplished in earlier studies. The same lineage tree is shown in **A'** with color-coded sub-lineages for each primary blast cell, pPGCs, and 8ML. The same color codes are used in the *Platynereis* larva cartoon in **B**. On the left in **B**, 4d lineage diagram is shown with primary blast cells labeled with the mesodermal region they contribute to. Note how the cells from the first two divisions of ML (pPGCs) end up next to the cells in the mesodermal posterior growth zone (MPGZ), which are born much later. For comparison, 4d (DM'') lineage diagram and primary blast cell lineages (again color-coded) are shown in **C** for the leech (a clitellate annelid) (*Cho et al., 2014*; *Gline et al., 2011*; *Rebscher, 2014*). The first six divisions of ML make cells that contribute to non-segmental mesoderm in anterior regions and the gut (both in dark gray). The following divisions make segmental primary blast cells, each or which contributes to the mesoderm of two consecutive hemisegments (for example, sm1 contributes to part of first segment and part of second segment mesoderm, in total making one segment-worth of mesoderm). Note that in the leech, PGCs (pink) arise in later cell divisions, as a subset within the segmental mesoderm population that originate from a primary blast cell (sm10, and sm12 through 22).
DOI: https://doi.org/10.7554/eLife.30463.048

tracing or cell ablation will establish whether *P. dumerilii* can produce (or regenerate) germline cells independently of the pPGCs discussed in this work.

4d lineage gives rise to the PGCs in other annelid species as well (*Rebscher, 2014*), but with some differences: in contrast to *P. dumerilii*, which segregates its pPGCs during the first two divisions of 4d (*Figure 8B*), PGCs arise much later from within the segmental mesoderm lineage in clitellates studied so far, as opposed to being set aside during the earlier divisions of ML/R. For example, in the leech, blast cells 16ml/r and 18 to 23ml/r (or sm10 and sm12-22 in the leech nomenclature) give rise to both segmental mesoderm and the PGCs within the associated segments (*Cho et al., 2014*; *Kang et al., 2002*) (*Figure 8C*). Cells produced by the first six divisions of 4d's daughters contribute mostly to gut endoderm, to head and pharynx musculature, but not to the germline. Likewise, in *Tubifex*, PGCs originate from within the progeny of 10ml/r and 11ml/r (*Kitamura and Shimizu, 2000*; *Niwa et al., 2013*; *Oyama and Shimizu, 2007*). In other spiralians such as mollusks, pPGCs are segregated from the 4d lineage during relatively early divisions (*Rebscher, 2014*), for example 2ml/r in the slipper snail (*Lyons et al., 2012*). However, the lineage that generates the pPGCs in the slipper snail also gives rise to mesodermal cell types. At the metazoan-wide level, it has been suggested that late epigenetic specification inside trunk mesoderm is the ancestral process and that early specification, often linked to the existence of a 'germ plasm', has evolved independently multiple times in metazoans (*Extavour and Akam, 2003*). What role these different segregation and specification patterns play, and whether they have implications in the ability to regenerate germ cells at later stages of development and in adulthood in a given species need to be further investigated.

## Formation of segmental and growth zone mesoderm in *P. dumerilii*

We have established, for the first time, the early pattern of segmental mesoderm formation in *P. dumerilii* (*Figure 4*, and *Figure 8* for summary). We found that during embryogenesis, segmental precursor cell segregation is similar to that in the leech or other clitellate embryogeneses. One important difference is that, only four segments (one anterior 'cryptic' segment and three bristle-bearing larval trunk segments) are formed during *P. dumerilii* embryonic/larval development. This is in sharp contrast to the 32 segments formed during leech embryogenesis and to the numerous segments formed in the other non-leech clitellates (*Anderson, 1973a*; *Balavoine, 2014*; *Goto et al., 1999b*; *Weisblat and Shankland, 1985*; *Zackson, 1982*). Nevertheless, as in clitellates, pairs of mesodermal precursors (called primary blast cells (PBCs)) are sequentially produced in an anterior to posterior progression, and correspond to a mesodermal segment of the larva. The segregation of the series of segmental PBCs in *P. dumerilii* (from 5ml/r to 8ml/r) is preceded by two pairs of precursors (3ml/r and 4ml/r) that divided only a few times during our analysis and probably contribute to the head mesoderm. These two pairs of anterior head mesoderm precursors may be related to the six pairs of em precursors known in the leech that contribute to endoderm and pharynx musculature (*Figure 8C*) (*Gline et al., 2011*).

We were not able to determine the precise tissue-type contribution of each mesodermal PBC because our lineage analysis ends at the beginning of late trochophore stage, before cell differentiation takes place. Additional imaging that extends further in development suggests that segmental blocks of mesodermal cells contributed mostly to a single hemisegment (*Figure 4—figure supplement 3*, *Figure 4—video 17*). However, we cannot exclude the possibility that each clonal block will eventually contribute complementary sets of different tissue types to two or more contiguous segments in the juveniles, as they do in the leech. Future studies will also determine whether these precursors give rise to a few neuronal cells (as in the clitellates), as well as other tissues such as the gut endoderm, and possibly pygidium (the non-segmental posterior end) as it has been previously suggested (*Starunov et al., 2015*).

Another important difference between *P. dumerilii* and clitellate annelids resides in the size of the mesodermal teloblasts and their evolution through embryogenesis. In the leech and other clitellates, M teloblasts are huge cells that undergo a long series of very asymmetric divisions, budding off a series of much smaller mesodermal PBCs, before they become 'exhausted' in terms of cytoplasmic volume after they produce the most posterior segments (*Shimizu, 1982*; *Weisblat and Kuo, 2014*). In *P. dumerilii*, correlating with the small number of segments produced during embryogenesis, M teloblasts are born with a medium size and appear to get quickly 'exhausted' through a short series of asymmetric divisions (*Figure 6*). It will be interesting to find out in the future, whether such M

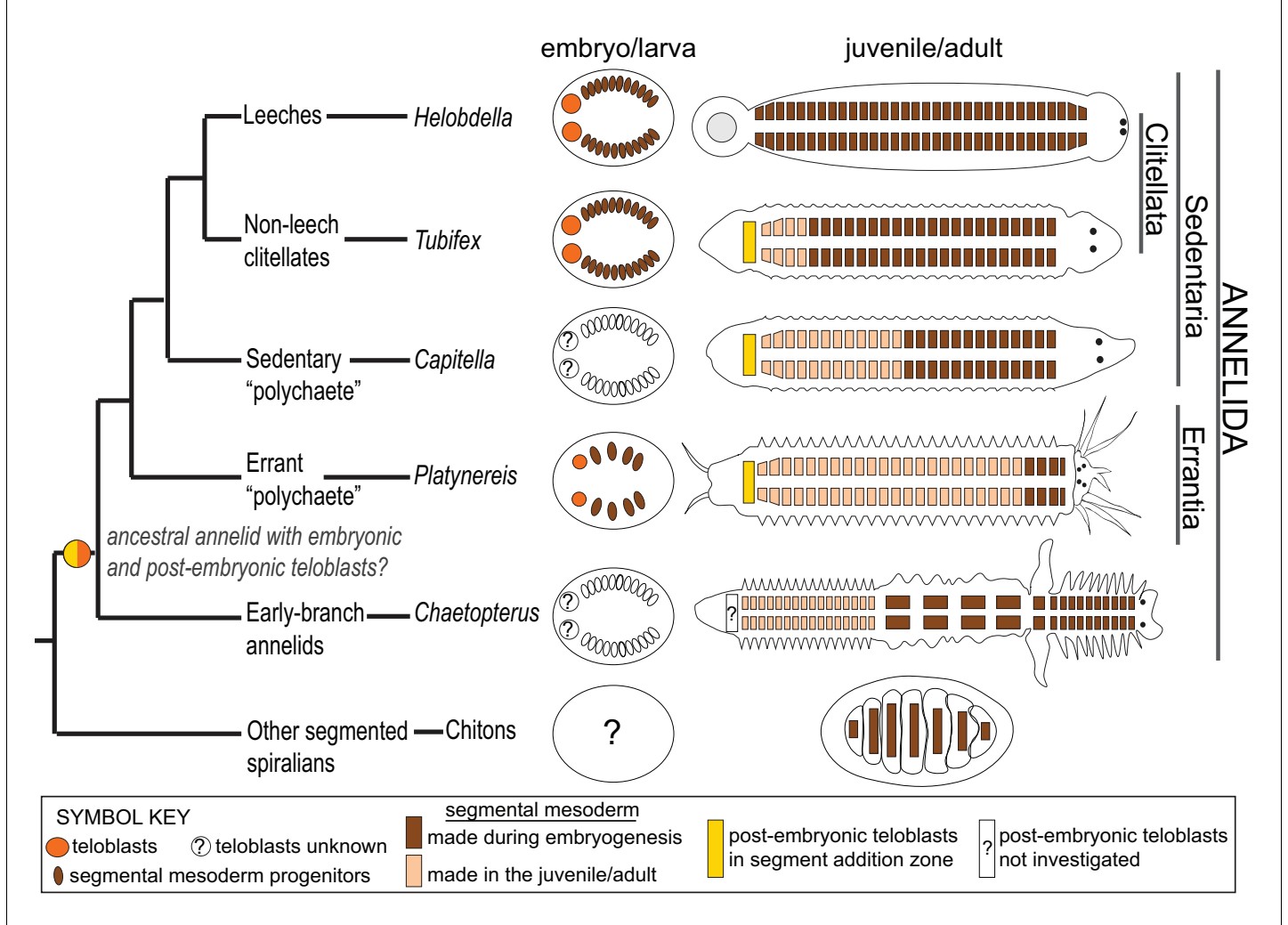

**Figure 9.** Phylogenetic distribution of embryonic and post-embryonic teloblasts in some annelids, and the relation to the number of segments made. A simplified annelid phylogeny (*Weigert and Bleidorn, 2016*) summarizes the main annelid species investigated for segment formation and teloblasts. The number of segments appearing during the pelagic larval phase (but patterned during embryogenesis) and the number of segments added posteriorly after the start of benthic life can vary considerably (*Balavoine, 2014*). In leeches, a fixed number of segments is made during direct embryonic development and none added after hatching. In non-leech clitellates (such as *Tubifex*), the embryos make a few tens of segments during embryogenesis and add many more after hatching. In non-clitellate annelids (Errantia, Sedentaria, and early branching), the number of larval segments is variable, and many more are added in post-larval development: In Nereidids (Errantia) such as *Platynereis*, only three larval segments develop, while in *Capitella* (Sedentaria), 13 larval segments (*Thamm and Seaver, 2008*) and in *Chaetopterus* (an early-branching annelid) 15 larval segments develop (*Seaver et al., 2001*). Embryonic and post-embryonic teloblasts differ in the way they have been evidenced. Embryonic teloblasts in the Clitellates (*Weisblat and Kuo, 2014*; *Goto et al., 1999a*; *Nakamoto et al., 2011*) and in *Platynereis* (this work) have been directly observed in live specimens. By contrast, post-embryonic teloblasts are inconspicuous cells that are identified only so far by their molecular stem cell signature (*Gazave et al., 2013*; *Dill and Seaver, 2008*; *Özpolat et al., 2016*). No direct observation of post-embryonic teloblast patterns of division is available so far. No direct evidence for embryonic teloblasts giving birth to post-embryonic teloblasts exists in any species so far to our knowledge. The ancestor of annelids presumably had both larval and post-larval segments, and thus it raises the question of the presence of both embryonic and post-embryonic teloblasts in this ancestral annelid, and even more broadly in other spiralians such as chitons (a type of segmented mollusk). Structures shown in the figure are color coded and explained in the 'symbol key'.

DOI: https://doi.org/10.7554/eLife.30463.049

teloblasts exists in non-clitellate annelids that form more segments during the embryonic/larval stage, such as *Capitella* which forms 13 larval segments (*Balavoine, 2014*; *Seaver, 2016*), and whether teloblast size evolved in proportion to the number of larval segments (*Figure 9*, also see discussion below).

We also found that after the progenitors of the four mesodermal segments have been produced, the remaining M teloblast cells in *P. dumerilii* gave rise to the precursors of mesodermal posterior growth zone (MPGZ), which is the likely source of the mesodermal component of the Segment Addition Zone (SAZ). Most annelids (but not leeches) continue growing from their posterior end by adding new segments throughout their lives (*Özpolat et al., 2016*) (*Figure 9*). The new tissues that form new segments originate from the SAZ which is a specific ring of small cells (presumably teloblast-like) within the posterior growth zone, and has mesodermal and ectodermal components (*Balavoine, 2014*). The SAZ expresses many Germline/Multipotency markers (*Gazave et al., 2013*; *Juliano et al., 2010*; *Özpolat et al., 2016*, *Özpolat et al., 2015*). Much is still unknown about the exact embryonic origins of the SAZ in annelids. Studies using bright-field microscopy, and DiI injections suggested that the mesodermal SAZ originates from the 4d blastomeres (*Ackermann et al., 2005*; *Anderson, 1973b*; *Rebscher et al., 2007*), but which cells within the 4d lineage make up the growth zone has not been identified to date. Our work successfully identifies 8ML/MR as the origin of the mesodermal SAZ in the larva. A slowly cycling small cluster of cells are made by 8ML/MR, located immediately anterior to the pPGCs, consistent with the previous studies that described this region as the Vasa(+) 'mesodermal posterior growth zone' (MPGZ) (*Rebscher et al., 2012*, *Rebscher et al., 2007*). Our imaging dataset is only until mid-/late-trochophore stage, thus we do not know whether all or a subset of these cells will produce the mesodermal SAZ in the later larval stages and juveniles. Thus, how these precursors go onto form the ring-like mesodermal SAZ and how they contribute to new tissues in juvenile worms remain to be determined.

This study in *P. dumerilii* helps putting different types of development in the annelids into a phylogenetic perspective (*Figure 9*). In the leeches, the totality of a species-specific number of segments are made during embryogenesis through the activity of large embryonic teloblasts. In contrast, in most marine annelids, the majority of segments are made during post-embryonic juvenile development, through the activity of inconspicuous posterior stem cells whose characteristics are still largely unknown. Earlier works in *P. dumerilii* (*de Rosa et al., 2005*; *Gazave et al., 2013*) have suggested that posterior stem cells in the juvenile may have a teloblast-like activity. We establish in this article that the mesodermal teloblasts which are very similar to the leech are indeed at play in the *P. dumerilii* embryo, contributing to the formation of the three larval segments. We also show that these embryonic teloblasts are most likely at the origin of the teloblast-like posterior stem cells of the juvenile (post-embryonic teloblasts), thus possibly establishing a cellular continuity in segment formation in *P. dumerilii*. In addition, it will be useful to determine if embryonic ectodermal teloblasts exist in *P. dumerilii* and whether they are at the origin of the ectodermal posterior stem cells as well. It is very likely that the ancestral life cycle in annelids involved these two phases of segment formation (embryonic/larval and post-embryonic) (*Figure 9*). The large teloblasts of Clitellate embryos would have evolved in the context of an acceleration of development, most or all segment formation happening in the embryonic stage (*Balavoine, 2014*). Meanwhile, parallels among all spiralians remain to be determined. It is legitimate to wonder whether embryonic teloblasts are present in mollusks having a form of mesodermal segmentation such as chitons or aplacophorans (*Scherholz et al., 2015*), or whether post-embryonic teloblasts are present in other posteriorly growing phyla, such as nemerteans. Future studies using high-resolution live imaging and genetic lineage-tracing methods on these exciting but understudied spiralian groups, as well as additional studies in the annelids will help answer the open questions surrounding teloblasts and their involvement in the evolution of different body plans.

## Materials and methods

**Key resources table**

| Reagent type (species) or resource | Designation | Source or reference | Identifiers | Additional information |
|---|---|---|---|---|
| Gene (*Platynereis dumerilii*) | Pdu-cdt1 | NA | GenBank No: MF614951 | From the full 2337 bp coding sequence, a 1946 bp part including the start site, and ending before the stop codon was amplified |

*Continued on next page*

*Continued*

| Reagent type (species) or resource | Designation | Source or reference | Identifiers | Additional information |
|---|---|---|---|---|
| Strain, strain background (*P. dumerilii*) | Wild Type | Institut Jacques Monod cultures | | |
| Recombinant DNA reagent | pCS2+ mVenus-cdt1 (aa1-147) | this paper | GenBank No: MF614950 | |
| Recombinant DNA reagent | pEXPTol2-H2A-mCherry | Other | | Plasmid was not specifically produced for this paper, and has been published before. The source plasmids used for the constructions of this plasmid was provided by Caren Norden Lab (MPI-CBG, Dresden, Germany). |
| Recombinant DNA reagent | pEXPTol2-EGFP-CAAX | Other | | Plasmid was not specifically produced for this paper, and has been published before. The source plasmids used for the constructions of this plasmid was provided by Caren Norden Lab (MPI-CBG, Dresden, Germany). |
| Antibody | Acetylated-tubulin | Sigma-Aldrich | T7451, RRID:AB_609894 | (1:500) |
| Sequence-based reagent | Pdu-cdt1- Forward Primer | this paper | | TTGTTTTCTTGTTGAGGTGGGATG |
| Sequence-based reagent | Pdu-cdt1- Reverse Primer | this paper | | GGAGATGACGAGACAGGCAG |
| Sequence-based reagent | pCS2+ vector backbone – Forward Primer | this paper | | CTAGAACTATAGTGTGTTGTATTACGT |
| Sequence-based reagent | pCS2+ vector backbone – Reverse Primer | this paper | | AGAGGCCTTGAATTCGAATCG |
| Commercial assay or kit | Gibson assembly master mix | NEB France | E2611S | |
| Commercial assay or kit | Click-iT EdU Alexa Fluor 647 Imaging Kit | Life Technologies | C10340 | |
| Commercial assay or kit | SP6 mMessage kit | Ambion, ThermoFisher | AM1340 | |
| Commercial assay or kit | MEGAclear kit | Ambion, ThermoFisher | AM1908 | RNA elution option two with the modifications - see Materials and methods for details |
| Commercial assay or kit | Proteinase K | Ambion, ThermoFisher | AM2548 | 20 µg/µl final concentration |
| Software, algorithm | Bitplane Imaris (Version 8.2) | | | |
| Software, algorithm | Onionizer | https://figshare.com/s/1c0dcd120d13deb888ba | | |

## Animal cultures and staging

*Platynereis dumerilii* embryos were obtained from lab cultures at the Institut Jacques Monod. Cultures are reared based on previous protocols (*Dorresteijn et al., 1993*) (available in English by Fischer and Dorresteijn on www.platynereis.de). Embryos and larvae are staged after *Fischer et al., 2010*. All staging is made at 18°C unless otherwise stated. For live time-lapse imaging, the imaging temperatures were typically higher (estimated 26–28°C). We provide a table with calculated developmental times and stages corresponding to each time point (*Table 1*).

## Gene cloning

*Pdu-cdt1* coding sequence was obtained from transcriptome databases by the Jékely and Arendt Labs (jekely-lab.tuebingen.mpg.de/blast; 4dx.embl.de/platy). An amino acid alignment against several species was carried out to confirm the sequence (MUSCLE alignment in GENEIOUS 6.1.8). From the full 2337 bp coding sequence, a 1946 bp part including the start site, and ending before the stop codon was amplified (*Pdu-cdt1*- Forward Primer: TTGTTTTCTTGTTGAGGTGGGATG; Reverse

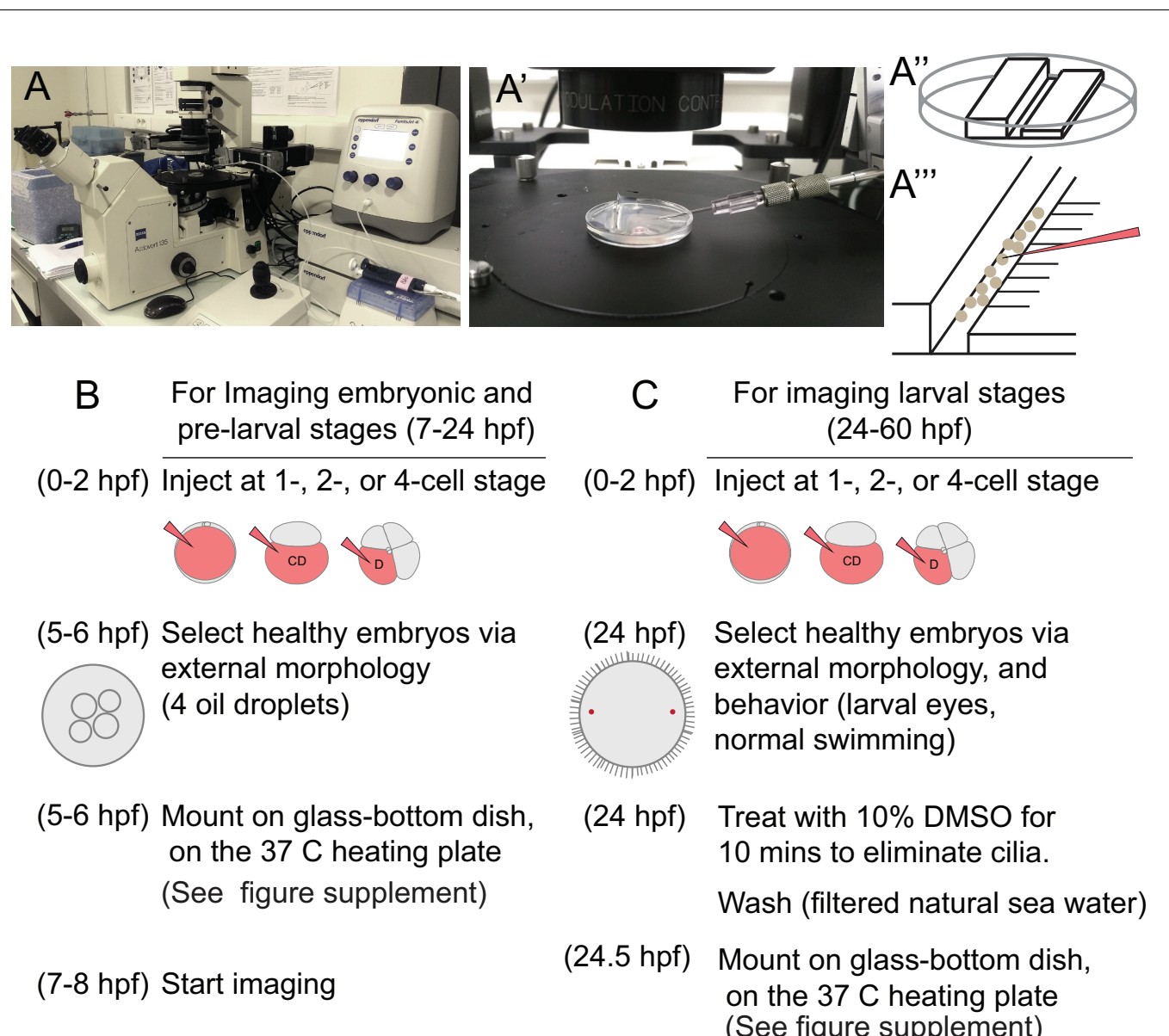

**Figure 10.** Injection setup and general experimental outline for imaging samples. (**A**) A Zeiss inverted scope with a gliding stage, coupled with the Eppendorf Femtojet microinjector and Eppendorf Transferman micromanipulation system were used for injections. Femtotip ready-to-use injection needles were back-filled with injection solution (**A'**). An agarose platform (**A''**) with a groove large enough to contain embryos was prepared using a plastic custom-made mold. Under a dissecting scope, small incisions were made to the right side of the groove (**A'''**), and these incisions were used to remove embryos from the needle by sliding the needle through them. The agarose platform was placed into a small petri dish lid and covered with filtered sea water before the embryos were transferred into the dish. In B and C, the steps of general experimental outline for live imaging samples at different stages are listed.

DOI: https://doi.org/10.7554/eLife.30463.050

The following figure supplement is available for figure 10:

**Figure supplement 1.** Heating plate setup for mounting samples in glass-bottom dishes.

DOI: https://doi.org/10.7554/eLife.30463.051

Primer: GGAGATGACGAGACAGGCAG). The PCR fragment was gel purified, cloned into TOPO-TA pCRII plasmid, and sequenced (GenBank No: MF614951).

## Construction of nuclear, membrane, and cell cycle constructs

### Nuclear and membrane constructs

We created the expression constructs pEXPTol2-EGFP-CAAX and pEXPTol2-H2A-mCherry by LR Recombination Reaction according to the manufacturer's instructions (Gateway Technology, Invitrogen,Carlsbad, CA). The individual plasmids used were the entry plasmids pENTR- p5E-CMV-SP6, pENTR-pME- EGFP-CAAX and pENTR-p3E-polyA or pENTR- p5E-CMV-SP6, pENTR-pME-H2A-mCherry, and pENTR-p3E-polyA and the destination plasmid pDestTol2pA2 (*Kwan et al., 2007*).

### Cell cycle construct

Using an alignment of *P. dumerilii* Cdt1 (*Pdu-cdt1*) with vertebrate Cdt1 amino acid sequences, we sub-cloned a C-terminal truncation of Cdt1 that is comparable to FUCCI constructs made by others previously (*Sakaue-Sawano et al., 2008*; *Sugiyama et al., 2009*), including the PIP degron, excluding Cdt1-Geminin interaction site and Cdt1-MCM2-7-binding site (*Figure 1A*). The truncated *Pdu-cdt1* sequence was placed at the 3' end of mVenus fluorescent protein. To ensure the stability of expressed or injected mRNA for the fused *mVenus-cdt1(aa1-147)* construct, we added a KOZAK sequence immediately upstream to the start codon of mVenus. The conserved 'CACC' KOZAK sequence from *H. sapiens* was used, because we could not determine a consensus sequence for *P. dumerilii*. Finally, several STOP codon sequences were added to the 3' end of the construct. This KOZAK-mVenus-cdt1(aa1-147) sequence was synthetized by a provider (IDTDNA gBlocks, Belgium). The synthetized fragment (200 ng) was resuspended in 20 µl nuclease-free water to a final concentration of 10 ng/µl, and it was cloned using Gibson reaction into pCS2+ expression vector which was amplified via PCR (pCS2+ vector backbone – Forward Primer: CTAGAACTATAGTGTGTTGTA TTACGT; Reverse Primer: AGAGGCCTTGAATTCGAATCG) (GenBank No: MF614950). For the Gibson reaction, Gibson assembly master mix (NEB, France, E2611S) and NEB 5-alpha Competent *E. coli* (NEB C2987H) were used, following the manufacturer's protocol.

## In vitro transcription of mRNA for microinjections

To express constructs in *P. dumerilii* embryos, we injected in vitro-transcribed mRNA into fertilized oocytes. To prepare the mRNA, each vector was linearized, gel-purified, and used for the in vitro transcription reaction (*HistoneH2A-mCherry: BglII/SP6; EGFP-caax: ClaI/SP6; mVenus-cdt1(aa1-147)*: Not-I/SP6). For in vitro transcription, SP6 mMessage kit from Ambion (AM1340) was used, and the manufacturer's protocol was followed until the end of DNase step (1 µl DNase 15 mins at 37°C). For purification of mRNA, MEGAclear kit from Ambion (AM1908) was used, following RNA elution option two with the following modifications: elution buffer was heated to 72°C, this warmed elution buffer was applied to filter cartridge containing mRNA, the tubes were kept at 72°C heated plate for 5 min before centrifuging for elution.

## Embryo preparation and microinjections

For preparing the embryos for microinjections, fertilized embryos were dejellied and their cuticle was softened via enzymatic degradation. To do this, fertilized embryos were transferred to a nylon 80 µm-mesh at about 1 hr-post-fertilization (hpf), washed 10 times using filtered natural sea water to remove the jelly secretion, treated with 20 µg/µl Proteinase K (Ambion, AM2548) in 30 mL seawater for 1 min, then washed again 10 times, and transferred to a six-well plate.

For microinjections, an agarose injection platform (made of 1.5% agarose in filtered natural sea water) was prepared (*Lauri et al., 2014*) (*Figure 10A–A'''*). Injection solutions were prepared using the following final concentrations: *mVenus-cdt1(aa1-147)* – 25 ng/µl, *HistoneH2A-mCherry* – 75 ng/µl, EGFP-caax – 75 ng/µl, 0.07% phenol red (Sigma-Aldrich, St. Louis, MO, P0290, 0.5% stock). Microinjections were carried out using an inverted Zeiss Axiovert 135 scope, Eppendorf Femtojet 4i, and Eppendorf Transferman Micromanipulator (*Figure 10A*). Micro injection needle (Sterile Eppendorf Femtotips II (930000043) was angled at about 15–20 degrees (*Figure 10A'*), and was connected to an Eppendorf universal capillary holder (920-00-739-2) with adapter (no: 5) for Femtotips. Femtojet pressure settings varied depending on the needle opening size, approximately in the following

ranges: pi[hPa]=200–300 psi; pc[hPa]=30–300 psi; ti = 1–4 s (pi: injection pressure, pc: back pressure, ti: injection time). Embryos were injected at one-cell stage, two-cell stage (into CD half), and four-cell stage (into D quadrant). Injected embryos were transferred back to the six-well plate and were kept at 18°C incubator until mounting.

## Mounting samples for imaging

For all mounts, we used glass-bottom dishes (MatTek Corporation 35 mm dish, P35G-0–10 C), as this method enabled keeping embryos and larvae in sea water which could be easily renewed if desired. As a result, samples were healthy after mounting during extended periods of imaging, and even after imaging they continued to grow in the agarose. Mounting was carried out on a warm plate to keep low-melting agarose from solidifying while allowing enough time for rotating the samples to the desired position using an eyelash tool (*Figure 10—figure supplement 1*). Low melting agarose (Invitrogen, France, 16520050) was prepared as 1% solution in filtered natural sea water by microwaving, and 500 µl aliquots were kept in −20°C until use. These tubes were thawed in a water bath at 65°C right before mounting, and kept in dry heating block at 37°C once thawed.

Injected embryos and larvae can be checked for normal development based on a number of criteria observed at different stages of development. For injected embryos to be imaged early (around 7–8 hpf at 18°C developmental time), we identified those injected samples without any visible deformations, clear animal-vegetal polarization, and four oil droplets with typical appearance at this stage (*Fischer et al., 2010*). For samples to be imaged after ciliary band development (15 hpf or later, at 18°C developmental time), we looked for normal ciliary bands, normal swimming behavior, chaetal sacs with developing bristles, and larval eyes. We picked healthy samples based on these criteria, as well as the samples with optimal fluorescence signal strength (which varies based on the amount of mRNA injected), and appropriate angle for mesoderm imaging after mounting.

Depending on the developmental stage to be imaged, samples had to be mounted before or after the ciliary band formation and the start of swimming movements. Mounting embryos or larvae with functional cilia will result in the specimens rotating in their agarose spherical lodge. For stages after the cilia form, immobilization requires deciliation at the time of mounting. We thus carried out imaging for both pre-hatching (embryonic and pre-larval) and post-hatching (larval) stages using two different techniques:

### Mounting technique 1 - Mounting before ciliary band formation (*Figure 10B*)

Cilia do not appear before 15 hpf (18°C). We observed that specimens mounted in low-melting agarose before 15 hpf (usually around 5–6 hpf) usually will not move even after the ciliary bands develop, because the growing cilia are impaired by the agarose covering. These specimens were simply placed onto the glass coverslip, excess sea water was removed, and low-melting agarose was added to cover the sample.

### Mounting technique 2 – Mounting after ciliary bands are formed (*Figure 10C*)

If specimens are to be imaged after the ciliary band formation, we found that incubating the larvae for 10 min in a small volume of 10% DMSO/90% seawater, followed by sudden flushing with fresh seawater caused clumping and breakage of the cilia, thus immobilizing the specimens temporarily. Embryos and larvae can then be mounted in low-melting agarose in the next ten minutes before the cilia start to grow back. These DMSO-treated larvae recover completely and grow without abnormalities. Mounting injected samples during larval stages may be desirable when a specific and precise orientation of the sample is required starting at a specific stage.

## EdU cell proliferation analysis and immunohistochemistry

### EdU assay on injected samples

Embryos were injected at one-, two-, or four-cell stage with *HistoneH2A-mCherry* and *mVenus-cdt1 (aa1-147)* solution mix, and were raised until 12 hpf. At 12 hpf, they were treated with 5 µM EdU in natural sea water for 3 min, rinsed quickly and fixed in 4% PFA for 1 hr. After fixation, they were washed in 1X PBS a few times and were processed for the EdU reaction next day, using the Click-iT

EdU Alexa Fluor 647 Imaging Kit (Life Technologies, C10340) kit following the manufacturer's protocol, except EdU reaction was carried out for 45 min.

## EdU assay and Immunohistochemistry

These experiments were done according to previously-published protocols (*Asadulina et al., 2012*; *Gazave et al., 2013*). For EdU assays, embryos or larvae were incubated in EdU (incubation conditions same as above) for the desired time period. Then EdU solution was washed by transferring the samples several times through sea water. The samples were incubated to grow until the desired stage and fixed in 4% PFA for 2 hr. The fixed samples were either processed just for EdU reaction, or additional immunohistochemistry was carried out (primary antibody: Acetylated-tubulin (1:500) (Sigma-Aldrich, T7451), secondary antibody: anti-mouse Alexa Fluor 488 (Molecular Probes, Life Technologies, Eugene, OR, 44408S), and DAPI (1:1000 from stock concentration of 1 mg/ml, overnight). EdU reaction was developed on the final day of immunohistochemistry protocol.

## Imaging

Immunohistochemistry and EdU reaction samples were mounted in 33% TDE (*Asadulina et al., 2012*) using standard slides and coverslips, and were imaged using a Zeiss LSM710 confocal scope. For the injected samples treated with EdU, endogenous mCherry and mVenus signals (expressed from the injected mRNA) are detectable in 'quick-fixed' specimens (see above), thus precluding the need for antibody detection.

For live imaging, a Zeiss LSM780 confocal scope was used. In the samples coinjected with both nuclear and membrane constructs, we took advantage of mVenus (YFP) being nuclear and EGFP being strictly localized to the cell membrane, and imaged the two fluorophores together, using single laser (514) for excitation. This allowed quick simultaneous acquisition of all three fluorophores decreasing phototoxicity due to laser exposure. Fluorescent protein production from the injected mRNA started becoming detectable around 5–6 hr after injections, thus imaging could not be carried out earlier. For the samples imaged for cell lineage analysis (Samples A, B, C in *Figure 5—figure supplement 1*), 1.13 μm step size and 60–70 μm total stack thickness was used, representing the proximal embryonic hemisphere roughly. The 4D Z-stack for Sample A has been uploaded online as a dataset and can be accessed at https://doi.org/10.5281/zenodo.1063531.

## Image analysis and processing

EdU assays, and immunostaining stacks were analyzed and exported using Fiji release of ImageJ (*Schindelin et al., 2012*). Nuclear signal counts were done manually in Fiji for Edu assays on injected samples. For live imaging datasets, 3D-projected stacks were exported as videos also using Fiji, except cell lineage tracking (see below). Time frame, and frequency of imaging is indicated for each video specifically. Annotated videos were edited using Adobe Premiere Pro CC 2017, and video encoding was adjusted to web-viewing using Hanbrake.

To better represent graphically an embryo that is spherical in shape, or parts of the embryo that are laid out in a curved manner, we wrote a Fiji script called 'Onionizer'. (The onionized images appear in *Figure 1—figure supplement 2C*, and *Figure 4—figure supplement 3*.) This is essentially a tool that allows in silico flattening of the embryo at various tissue depths. Concretely, the script extracts the intersection of the 3D stacks and a series of near spherical surfaces located at increasing depth from the tissue surface. The intersections are then 2D projected as if successive 'onion layers' of the sample were flattened. The stack (x; y; z) is replaced by a (x; y; ol) stack where 'ol' represent onion layers of increasing depth. This allows flattening of ectodermal tissues for low 'ol' values and mesodermal for higher 'ol' values. The FIJI script is available on FigShare.com (https://figshare.com/s/1c0dcd120d13deb888ba).

All the cell lineage and cell cycle analyses were done using Bitplane Imaris (Version 8.2). Three different samples from two independent injection batches were used for lineage analysis, and these are letter coded throughout the manuscript: Sample A, Sample B, and Sample C. In the *Figures 3–7*, data from Sample A is shown (except in *Figure 5B–B''* Sample B is shown), and data from Samples B and C are presented in the figure supplements.

## Imaris – cell lineage

Zeiss confocal stacks were uploaded to Imaris, and cell lineage tracing was done manually using spots feature, volume and slice views in Imaris (as opposed to automated tracing, which does not work well when cell and nuclei sizes change drastically, as is the case in *P. dumerilii* embryos). Based on the manual curation and connection of spots across time frames, a cell lineage tree is automatically created by Imaris. Different color codes were assigned to different lineages (such as blue, orange, and pink). Descendants of particular blastomeres were selected using a different spot color (yellow) for highlighting the clonal regions, and time-lapse videos including these highlighted spots were exported using Imaris.

## Imaris – cell cycle

Spots (each sphere marking a cell in Imaris dataset) can be used for measuring fluorescence intensity for the pixels that fall into that spot's volume. We used this feature for obtaining a quantitation of the cell cycle construct. To do this, spot size and position were manually adjusted so that each spot was slightly smaller than the nucleus signal, and was positioned to be completely inside the nucleus. Fluorescence intensity mean calculation for the mVenus channel was extracted for different cell lineages. During mitosis (when cell nucleus disappeared) the spot was positioned roughly to contain chromosomes. Criteria for choosing lineages to compare was as follows: we picked always the larger of the two sister cells after a division, and if the division was equal, we picked the one more posterior/closer to the Mesoblast. Cell cycle graphs were drawn in Excel and were modified in Adobe Illustrator CC 2017.

## Acknowledgements

We thank the Balavoine Lab Members, Vervoort Lab Members, and Ryan Null for helpful discussions and feedback on the manuscript. We thank Solène Song for help with the 'onionization' code. We acknowledge the Company of Biologists for the travel award to BDO, which enabled a trip to Florian Raible's laboratory (MFPL, Vienna, Austria) for learning injections. We thank Karim Vadiwala (Raible Lab) for his assistance during this training period. We acknowledge the ImagoSeine facility (member of the France BioImaging infrastructure supported by the French National Research Agency, ANR-10-INSB-04, 'Investments of the future') and its experienced staff for their assistance in imaging and image analysis. We also thank Caren Norden Lab (MPI-CBG, Dresden, Germany) for providing the plasmids for the construction of EGFP-caax and HistoneH2A-mCherry vectors. Finally, we thank the editors and reviewers for their comments that helped improve this manuscript.

## Additional information

### Funding

| Funder | Grant reference number | Author |
| --- | --- | --- |
| Labex | No.ANR-11-LABX-0071 | Michel Vervoort<br>Guillaume Balavoine |
| Agence Nationale de la Recherche | METAMERE no. ANR-12-BSV2-0021 | Michel Vervoort<br>Guillaume Balavoine |
| Agence Nationale de la Recherche | TELOBLAST no. ANR-16-CE91-0007 | B Duygu Özpolat<br>Michel Vervoort<br>Guillaume Balavoine |

The funders had no role in study design, data collection and interpretation, or the decision to submit the work for publication.

### Author contributions

B Duygu Özpolat, Conceptualization, Resources, Data curation, Formal analysis, Validation, Investigation, Visualization, Methodology, Writing—original draft, Project administration, Writing—review and editing; Mette Handberg-Thorsager, Resources, Methodology, Writing—review and editing; Michel Vervoort, Conceptualization, Supervision, Funding acquisition, Methodology, Writing—review

and editing; Guillaume Balavoine, Conceptualization, Data curation, Software, Supervision, Funding acquisition, Validation, Methodology, Project administration, Writing—review and editing

### Author ORCIDs
B Duygu Özpolat  http://orcid.org/0000-0002-1900-965X
Mette Handberg-Thorsager  http://orcid.org/0000-0002-3908-7233
Guillaume Balavoine  https://orcid.org/0000-0003-0880-1331

### Decision letter and Author response
Decision letter https://doi.org/10.7554/eLife.30463.056
Author response https://doi.org/10.7554/eLife.30463.057

## Additional files

### Supplementary files
• Transparent reporting form
DOI: https://doi.org/10.7554/eLife.30463.052

### Major datasets
The following dataset was generated:

| Author(s) | Year | Dataset title | Dataset URL | Database, license, and accessibility information |
|---|---|---|---|---|
| Ozpolat BD, Handberg-Thorsager M, Vervoort M, Balavoine G | 2017 | Sample A - Z-stacks | https://doi.org/10.5281/zenodo.1063531 | Available at Zenodo under a Creative Commons Attribution-Non Commercial-No Derivatives license |

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
