## [Decision Letter]

Thank you for submitting your article "Cell lineage and cell cycling analysis of the 4d lineage using live imaging in the marine annelid *Platynereis dumerilii*" for consideration by *eLife*. Your article has been favorably evaluated by K VijayRaghavan (Senior Editor) and three reviewers, one of whom is a member of our Board of Reviewing Editors. The reviewers have opted to remain anonymous.

The reviewers have discussed the reviews with one another and the Reviewing Editor has drafted this decision to help you prepare a revised submission.

Summary:

Ozpolat and colleagues report their elegant in vivo time-lapse imaging of early embryogenesis of the marine annelid *Platynereis dumerilli*, for which they generated a FUCCI cell cycle reporter based on *P. dumerilli* cdt1 gene and optimized methods for long-term microscopy for this species. The experimental focus is on the 4d lineage, which was previously shown to give rise to mesodermal segments and primordial germ cells but the underlying cellular mechanisms were unknown. Leveraging these technological advances to extend imaging and 4d lineage tracing the authors demonstrate early segregation of PGCs in that linage, following by their clustering and cell cycle quiescence. Moreover, the traced cells divide to form several segment rudiments, not as many as in leeches but using a similar teloblast asymmetric division mechanism. This is a novel observation, along with the extension of the lineage analysis by up to 8 cell generations. In addition, the authors report identification of the embryonic origins of mesodermal posterior growth zone and report different cell cycling properties for differentially-fated progenies. The importance of this work lies in that provides high resolution in vivo analyses of early development that extend earlier studies. The methods will be valuable for future studies of this and other marine species. Moreover, this work reveals a lineage separation that is similar to the well-studied teloblasts, like leaches. This is, evolutionarily, a surprising result, since it was thought that leeches were highly derived. Conversely, this work implies that leeches simply took a common feature of segmentation that is found throughout the phylum and amplified it.

Essential revisions:

1) The time interval in time-lapse analyses of several minutes is relatively long questioning the lineage relationships established by Imaris. Therefore, a manual validation of critical lineages to corroborate automatic analysis is essential.

2) Pdu FUCCI system to be better characterized, and its utility for this study better justified. Why Pdu FUCCI system is also labelling G2? Currently, it is not clear whether it provides additional information over the live imaging and tracing.

3) The paper needs to be presented better in terms of figure numbering, labeling and how figures are incorporated in the logical flow of the manuscript. Moreover, whereas the lineage tracing is the most important aspect of this work it is presented as an add on. Finally, the authors appropriately acknowledge that even their extended lineage tracing does not allow them to trace the lineages until the morphologic segments are apparent. It would be important to more clearly present the argument that the linear arrangements of specific sub-lineages represent segmental anlagen.

4) Videos 4.1 and 4.2 could not be converted. Videos 5.3 and 5.4 are missing.

[Editors' note: further revisions were requested prior to acceptance, as described below.]

Thank you for resubmitting your work entitled "Cell lineage and cell cycling analysis of the 4d micromere using live imaging in the marine annelid *Platynereis dumerilii*" for further consideration at *eLife*. Your revised article has been favorably evaluated by K VijayRaghavan (Senior Editor) and a Reviewing Editor.

The manuscript has been improved and in principle will be suitable for publication. However, there are some remaining issues that need to be addressed before acceptance, as outlined below:

The authors state that they have added to the figures labels explaining what fluorescent probes/proteins are shown in the images. Yet, it is not clear what we are looking at on Figure 3—figure supplement 2?

Same for Figure 4. The color key if the same should be also provided on the pages with figures supplemental to Figure 4.

Introduction, fourth paragraph: "PGCs" abbreviation should be explained.

Subsection “Establishment of a work flow for long time-lapse live imaging and cell lineage tracing in marine embryos and larvae”, second paragraph: "…each image stack acquisition (…) was acquired" revise "each image stack was acquired".

Subsection “Live imaging the 4d (M) lineage in *P. dumerilii* with a live-cell cycle reporter”: the sentence "As a result, behavior of specific lineages with detailed cell cycling characteristics at single-cell resolution was not possible," should be revised "behavior.…could not be discerned/observed” etc.

---

## [Author Response]

Essential revisions:1) The time interval in time-lapse analyses of several minutes is relatively long questioning the lineage relationships established by Imaris. Therefore, a manual validation of critical lineages to corroborate automatic analysis is essential.

All of the lineage relationships reported in this manuscript were curated manually and not via automatic algorithms in Imaris. This was already stated in Materials and methods but we added in several places in the text that the lineages were curated manually (subsection “Each mesodermal hemisegment is the progeny of individual blast cells produced in successive divisions”, first two paragraphs, subsection “Divisions generating segmental mesoderm precursors happen in quick succession, then cell cycling slows down significantly”, last paragraph).

2) Pdu FUCCI system to be better characterized, and its utility for this study better justified. Why Pdu FUCCI system is also labelling G2?

This was initially explained in the Discussion, but in order to make it more accessible early on in the manuscript, we have moved the information to Results and emphasized it in the last paragraph of the subsection “In *P. dumerilii,* mVenus-Cdt1(aa1-147) reporter”. In brief, the current cycling pattern obtained actually matches the endogenous patterns reported in other organisms, thus it is not surprising that the late G2 phase is labeled. In addition, for the purposes of the work in this manuscript, the cycling pattern observed was sufficiently useful. However, we are currently working on developing additional live-cell cycle reporters that will more precisely match different phases of the cell cycle. While we believe this ongoing work is out of the scope of this manuscript, it is worth noting that as a part of these efforts we have tested several different constructs obtained from the authors of the original FUCCI systems based on zebrafish and human sequences (Sakaue-Sawano et al., 2008) and did not observe fluorescence.

Currently, it is not clear whether it provides additional information over the live imaging and tracing.

Pdu-FUCCI was indeed crucial for cell lineage analysis carried out in this manuscript, as it provided additional resolution into cell cycle, making it feasible to trace lineages manually under the time and image resolution constraints we had. To clarify this, we added a section at the beginning of Results.

To our knowledge, with current algorithms available via different cell lineage-tracing platforms (such as Imaris, or the Fiji plugin Mamut), it is not possible to carry out an automated lineage analysis for embryos like *P. dumerilii*’s which have significantly variable cell sizes and nuclei. Most tracing algorithms rely on more or less uniform nuclei size for confidently tracking them from a time point to the next, such as it has been done in the early *C. elegans* and *Drosophila* embryos (Tomer et al., 2012; Wu et al., 2011). Consequently, lineage tracing cell in embryos such as *P. dumerilii*’s still rely on manual curation, especially at stages of development we have attempted in this work. During this manual curation of lineages, the researcher has to mark the nuclei one by one, from one frame to the next. We have optimized the imaging conditions for healthy-developing embryos, and in doing so, we had to give up better time- and image- resolution (therefore decreasing the time embryos were exposed to laser light for imaging). As a result, each frame had to be carefully examined to make sure to pick the right nuclei as the daughters of a dividing cell. The cell cycle reporter was critical at this stage, as it enabled observation of a particular cell’s progression in cell cycle, therefore making it easier to differentiate it from the surrounding cells, and predict if it is likely to divide. For example, if a nucleus being traced was red (meaning it is in early G2 phase) and it became orange/yellow (the brief period in late G2 when cells upregulate the cell cycle construct right before division), it was possible to predict the cell will divide soon (possibly in the next frame). This layer of extra information was crucial in accurate manual curation of lineages, especially as the cells progressively became smaller, division after division.

Of note, for this study, we did not have access to SPIM (i.e. light sheet microscopy), which would have allowed higher time resolution without deleterious effects. We used the cell cycle reporter to complement live imaging conditions with the technology available to us. We believe that our work may encourage others who have access to only confocal imaging. As data processing, visualization, and analysis remain a challenge in light sheet (Girstmair et al., 2016; Shah et al., 2017) scanning confocal imaging continues to be useful for live-imaging and lineage-tracing at single cell level, with the right combination of techniques.

3) The paper needs to be presented better in terms of figure numbering, labeling and how figures are incorporated in the logical flow of the manuscript. Moreover, whereas the lineage tracing is the most important aspect of this work it is presented as an add on.

To emphasize cell lineage analysis we put Figure 2 as a figure supplement to Figure 1 (new numbering: Figure 1—figure supplement 2) and removed the section from the main text. We also shortened some of the cell cycle discussion and instead of having this as the final discussion of the manuscript, we placed this part earlier in the Discussion (subsection “Cell cycle patterns of the 4d lineage”).

To emphasize the importance of the lineage analysis in a larger evo-devo picture, we have added an additional summary figure (Figure 9) that shows phylogenetic distribution of our current knowledge on teloblasts across annelids (and spiralians), and we conclude the manuscript with a discussion of these concepts (subsection “Formation of segmental and growth zone mesoderm in *P. dumerilii*”, last paragraph).

Finally, the authors appropriately acknowledge that even their extended lineage tracing does not allow them to trace the lineages until the morphologic segments are apparent. It would be important to more clearly present the argument that the linear arrangements of specific sub-lineages represent segmental anlagen.

To address this point we have made several additions to the manuscript (subsection “Each mesodermal hemisegment is the progeny of individual blast cells produced in successive divisions”, fourth paragraph):

1) We have added a figure supplement showing how the distinct mesodermal clonal cell clusters (i.e. one sub-lineage) align with the chaetal sacs of the corresponding hemisegment (Figure 3—figure supplement 2). This supplementary figure also demonstrates the position of the mesodermal clusters in relation to chaetal sacs more clearly.

2) To demonstrate how these early mesodermal clusters eventually encase the chaetal sacs in each hemisegment, and the cells mainly stay in their designated segmental anlage, we added a new time-lapse video and a corresponding figure (Figure 4—figure supplement 3, Video 8).

4) Videos 4.1 and 4.2 could not be converted. Videos 5.3 and 5.4 are missing.

5.3 and 5.4 do not exist, thus we removed these from the text. We uploaded new versions for all the other videos in correct formatting as required by *eLife*.

Bibliography:

Girstmair, J., Zakrzewski, A., Lapraz, F., Handberg-Thorsager, M., Tomancak, P., Pitrone, P.G., Simpson, F., Telford, M.J., 2016. Light-sheet microscopy for everyone? Experience of building an OpenSPIM to study flatworm development. BMC Dev. Biol. 16, 22. doi:10.1186/s12861-016-0122-0

Shah, G., Weber, M., Huisken, J., 2017. Light Sheet Microscopy, in: Fluorescence Microscopy. Wiley-VCH Verlag GmbH & Co. KGaA, Weinheim, Germany, pp. 243–265. doi:10.1002/9783527687732.ch7

Tomer, R., Khairy, K., Amat, F., Keller, P.J., 2012. Quantitative high-speed imaging of entire developing embryos with simultaneous multiview light-sheet microscopy. Nat. Methods 9, 755–763. doi:10.1038/nmeth.2062

Wu, Y., Ghitani, A., Christensen, R., Santella, A., Du, Z., Rondeau, G., Bao, Z., Colon-Ramos, D., Shroff, H., 2011. Inverted selective plane illumination microscopy (iSPIM) enables coupled cell identity lineaging and neurodevelopmental imaging in Caenorhabditis elegans. Proc. Natl. Acad. Sci. 108, 17708–17713. doi:10.1073/pnas.1108494108

[Editors' note: further revisions were requested prior to acceptance, as described below.]

The manuscript has been improved and in principle will be suitable for publication. However, there are some remaining issues that need to be addressed before acceptance, as outlined below:The authors state that they have added to the figures labels explaining what fluorescent probes/proteins are shown in the images. Yet, it is not clear what we are looking at on Figure 3—figure supplement 2?Same for Figure 4. The color key if the same should be also provided on the pages with figures supplemental to Figure 4.Introduction, fourth paragraph: "PGCs" abbreviation should be explained.Subsection “Establishment of a work flow for long time-lapse live imaging and cell lineage tracing in marine embryos and larvae”, second paragraph: "…each image stack acquisition (…) was acquired" revise "each image stack was acquired".Subsection “Live imaging the 4d (M) lineage in P. dumerilii with a live-cell cycle reporter”: the sentence "As a result, behavior of specific lineages with detailed cell cycling characteristics at single-cell resolution was not possible," should be revised "behavior.…could not be discerned/observed” etc.

We addressed the requests by making the following changes: We have added fluorophore information to all the relevant figures, and if this wasn’t possible to add this information in the figure due to space limitations, we made sure to add the information into the legend. (The following figures and/or their legends were edited: Figure 2, Figure 3, Figure 3—figure supplement 1 and Figure 3—figure supplement 2, Figure 4, Figure 4—figure supplement 1 and Figure 4—figure supplement 2, Figure 5). We have also added a DOI link to the complete Z-stack dataset uploaded to a cloud server (Zenodo) for sharing this dataset with other researchers. This information can be found at the end of the subsection “Imaging”.